# Engineered bacterial outer membrane vesicles encapsulating oncolytic adenoviruses enhance the efficacy of cancer virotherapy by augmenting tumor cell autophagy

Weiyue Ban[1,5], Mengchi Sun[2,5], Hanwei Huang[3,5], Wanxu Huang ⓘ[1,5], Siwei Pan[3], Pengfei Liu[3], Bingwu Li[1], Zhenguo Cheng[4], Zhonggui He[1], Funan Liu ⓘ[1,3] ✉ & Jin Sun ⓘ[1] ✉

Oncolytic adenovirus (Ad) infection promotes intracellular autophagy in tumors. This could kill cancer cells and contribute to Ads-mediated anticancer immunity. However, the low intratumoral content of intravenously delivered Ads could be insufficient to efficiently activate tumor over-autophagy. Herein, we report bacterial outer membrane vesicles (OMVs)-encapsulating Ads as microbial nanocomposites that are engineered for autophagy-cascade-augmented immunotherapy. Biomineral shells cover the surface antigens of OMVs to slow their clearance during in vivo circulation, enhancing intratumoral accumulation. After entering tumor cells, there is excessive $H_2O_2$ accumulation through the catalytic effect of overexpressed pyranose oxidase ($P_2O$) from microbial nanocomposite. This increases oxidative stress levels and triggers tumor autophagy. The autophagy-induced autophagosomes further promote Ads replication in infected tumor cells, leading to Ads-overactivated autophagy. Moreover, OMVs are powerful immunostimulants for remolding the immunosuppressive tumor microenvironment, facilitating antitumor immune response in preclinical cancer models in female mice. Therefore, the present autophagy-cascade-boosted immunotherapeutic method can expand OVs-based immunotherapy.

An attractive immunotherapeutic strategy is oncolytic viral biotherapy against cancer. It could selectively kill cancer cells and activate the systemic immune response using oncolytic viruses (OVs)[1,2]. Oncolytic adenoviruses (Ads) are commonly employed OVs due to their safety and efficacy[3]. Unfortunately, the clinical outcomes of potential Ads are short of expectations with several aspects as their cause[4]. First, as immunogenic biological particles, Ads are rapidly inactivated and cleared after intravenous injection, causing poor enrichment efficiency

at the tumor site[5]. Most current Ads in clinical research are intratumorally administrated into superficial tumors, impotent for certain deep tumors, and minute tumor lesions[6,7]. Secondly, the replication ability of Ads is still limited even if they reach tumor regions. Ads-infected stromal cells are less likely to be permissive to Ads replication. In contrast, the rapid apoptosis of Ads-infected cancer cells would shorten the effective viral replication time, ultimately decreasing the number of Ads in the tumor[8]. In addition, another major hindrance to

**Fig. 1 | Schematic diagram.** The biomineralized microbial nanocomposite engineered from OVs for autophagy-cascade-augmented immunotherapy.

resisting Ads-based immunotherapy is the immunosuppressive tumor microenvironment (TME). TME immunosuppressive modulators, such as regulatory T cells (Tregs) and pro-tumoral factors, cause immune evasion and decrease the efficacy of oncolytic antitumor[9].

The Ads-mediated tumor regression is strongly related to virus-activated autophagy in tumor cells[10,11]. Autophagy is critical in pro-viral replication, cancer cell killing, and antitumor immune activation. Autophagy is generated in the infected cells after selectively internalizing oncolytic Ads using target cancer cells. On the one hand, the in-situ synthesis of autophagy-induced double-membrane-bound vesicles (autophagosomes) could serve as viral replication sites during Ads infection to increase virion production[12,13]. On the other hand, Ads-activated autophagy could trigger autophagic immunogenic cell death. It releases damage-associated molecular pattern molecules and tumor-associated antigens, activating antitumor immune responses[14]. However, autophagy is a double-edged sword in cancer therapy, and different autophagy levels could have different effects[15]. The Ads-activated autophagy level is moderate due to the above application status and the predicament of oncolytic Ads. Thus, it plays a cytoprotective role and helps tumor cells to resist Ads treatment, causing unsatisfactory clinical efficacy. Therefore, it is clear that timely and precise overactivated autophagy in tumor sites by Ads-based therapy is crucial.

The current study addressed the clinical obstacles of Ads and utilized the physiologic characteristics of Ads-mediated autophagy. Therefore, we propose a microbial nanocomposite concept by encapsulating OVs into biomineral calcium phosphate (CaP)-camouflaged and pyranose oxidase ($P_2O$)-engineered bacterial outer membrane vesicles (OMVs) to enable autophagy-cascade-augmented antitumor immunotherapy (Fig. 1). The CaP shell protects OVs from the removal of the innate immune system after intravenous injection. Thus, the clearance is circumvented during the in vivo circulation, enhancing intratumoral accumulation. Upon entering tumor cells, $H_2O_2$, a reactive oxygen species (ROS) molecule, is converted in situ from endogenous glucose using $P_2O$ engineered from microbial nanocomposite. Notably, ROS generation significantly enhances the intracellular level of autophagy, forming autophagosomes[16]. The induction of autophagosomes would improve viral replication. This could facilitate the virus-induced overactivated autophagy, leading to cancer cell death[12,13]. The microbial nanocomposite could augment autophagic oncolytic immunotherapy by leveraging TME remolding ability of OMVs[17,18]. We report an autophagy-cascade-augmented OVs immunotherapy platform that provides insight for clinical applications based on enhanced OVs-mediated cancer immunotherapy.

## Results

### Preparation and in vitro evaluation of the microbial nanocomposite

In this study, the Ads were encapsulated by the engineered OMVs extracted from $P_2O$-expressed *E. coli*. Dynamic light scattering (DLS) and transmission electron microscopy (TEM) results confirmed the

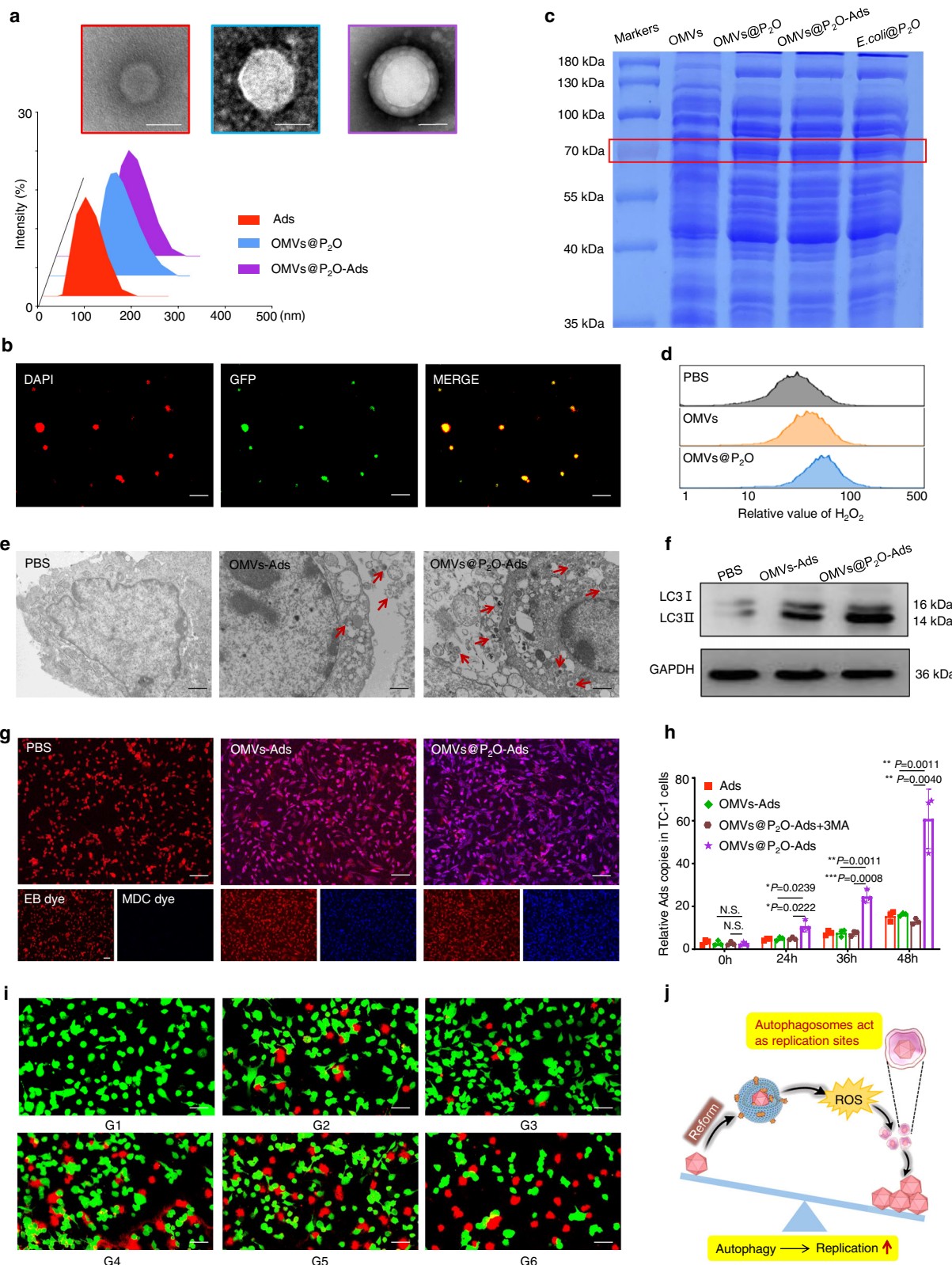

successful microbial nanocomposite preparation. They were named as OMVs@$P_2O$-Ads (Fig. 2a). OMVs@GFP-Ads were also constructed with the above protocol using engineered OMVs extracted from *E. coli* expressing the green fluorescent protein (GFP) marker. The confocal laser scanning microscopic (CLSM) images revealed the GFP-expressed OMVs shells around DAPI-labeled Ads (Fig. 2b). $P_2O$ expression of *E. coli*@$P_2O$, OMVs@$P_2O$, and the microbial nanocomposite was quantificationally assessed via

the biochemical colorimetric approach (Supplementary Figs. 1, 2). A functional curve helped determine the relative concentration of $P_2O$ within the microbial nanocomposite by calculating the absorbance of $P_2O$ under different concentrations (Supplementary Figs. 3, 4). Meanwhile, the sodium dodecyl sulfate-polyacrylamide gel electrophoresis (SDS-PAGE) method helped examine the existence of $P_2O$ within *E. coli*@$P_2O$, OMVs@$P_2O$, and the microbial nanocomposite (Fig. 2c).

**Fig. 2 | Preparation and in vitro evaluation of the microbial nanocomposite.** **a** TEM and size distribution images of Ads, OMVs@P$_2$O, and OMVs@P$_2$O-Ads. Scale bar=100 nm. **b** CLSM images of the microbial nanocomposite. Ads were stained with DAPI dye (red) and OMVs carried a GFP marker (green). Scale bar = 1 μm. **c** The expression of P$_2$O was investigated by the SDS-PAGE method. This experiments in (**a**–**c**) were repeated three times independently with similar results. **d** The ROS level assessment in TC-1 cells by flow cytometry. **e** TEM images of autophagosomes. Scale bar = 1 μm. **f** The expression of autophagy-related protein LC3-I and LC3-II by western bolt analyses. **g** CLSM images of autophagosomes. Cells were stained with EB dye (red) and autophagosomes were stained with MDC dye (blue). Scale bar = 50 μm. This experiments in (**e**–**g**) were repeated three times independently with similar results. **h** The Ads replication in TC-1 cells was quantified using real-time PCR at 0, 24, 36, and 48 h sequentially. 3MA is an autophagy inhibitor: 3-Methyladenine. Data are presented as mean ± SD ($n$ = 3 independent experiments). N.S. (No Significance) $P > 0.05$, *$P < 0.05$, **$P < 0.01$ and ***$P < 0.001$ by two-tailed Student's t-test. Source data are provided as a Source Data file. **i** Cytotoxicity of different formulations in TC-1 cells by CLSM. Living cells were stained with Calcein (green) and dead cells were stained with PI (red). Scale bar = 20 μm. This experiment was repeated three times independently with similar results. **j** Schematic diagram of bridging ROS with oncolytic Ads replication. *$p < 0.05$, **$p < 0.01$, ***$p < 0.001$, ****$p < 0.0001$ versus control. G1: PBS, G2: Ads, G3: OMVs, G4: OMVs@P$_2$O, G5: OMVs-Ads, G6: OMVs@P$_2$O-Ads.

The OMVs were extracted as the negative control from non-engineered *E. coli*. The ROS generation capacity of OMVs@P$_2$O and OMVs in a non-small cell lung (TC-1) cancer cell line was investigated using flow cytometry. As shown in Fig. 2d and Supplementary Fig. 5, the ROS production of OMVs@P$_2$O was higher than that of OMVs. Similar results of ROS production were obtained by CLSM (Supplementary Fig. 6). High levels of ROS could lead to severe oxidative stress, resulting in overactivated autophagy. The number of autophagosomes regarded as the intuitive index assessing the autophagy level was directly observed using TEM (Fig. 2e and Supplementary Fig. 7). LC3 is one of the most typical autophagy-related proteins. Once the autophagy is activated, LC3 is converted from LC3-I to LC3-II. Thus, the LC3-II/LC3-I ratio is utilized to determine the autophagy level using western blot[15]. As depicted in Fig. 2f and Supplementary Fig. 8, the microbial nanocomposite treatment led to an increased LC3-II/LC3-I ratio than the OMVs-Ads treated group. Besides, compared to PBS and OMVs-Ads groups, the microbial nanocomposite-treated group exhibited the abnormal over-accumulation of autophagosomes (Fig. 2g)[19]. These results indicated that the engineered OMVs-generated ROS could enhance autophagy and produce autophagy-induced autophagosomes. In addition, significantly improved Ads production could be observed in Ads-infected TC-1 cells after the microbial nanocomposite treatment, compared to Ads alone or OMVs-Ads (Fig. 2h). Autophagy-generated internal double-membrane-bound vesicles (autophagosomes) could be Ads replication sites within Ads-infected tumor cells. This effectively enhanced Ads replication through the autophagy pathway (Fig. 2j). They became more potent by autophagy-augmented Ads replicating the microbial nanocomposite candidates for in vitro antitumor evaluation. Therefore, the microbial nanocomposite had the strongest in vitro tumor cell-killing ability (Fig. 2i and Supplementary Figs. 9, 10).

## In vivo oncolytic efficacy of the microbial nanocomposite

We qualitatively explored the in vivo biodistribution of the microbial nanocomposite using intratumoral injection (i.t.). The TC-1-hCD46 xenograft tumor-bearing C57 female mice were intratumorally injected using the microbial nanocomposite with a fluorescent dye, DIR. In the in vivo spectral CT imaging system (IVIS), the fluorescence intensity was measured in mice at 0, 12, 24, and 36 h. Figure 3a and Supplementary Fig. 11 showed that the microbial nanocomposite was restricted to the treated tumors. Moreover, no fluorescence of the microbial nanocomposite could be observed in other organs of the treated mice. Then, we investigated the antitumor activity of the microbial nanocomposite against the TC-1-hCD46 xenograft tumor-bearing C57 female mice model. Tumor-bearing mice were divided into six groups sequentially receiving i.t. injections of PBS, Ads, OMVs, OMVs@P$_2$O, OMVs-Ads, and the microbial nanocomposite (G1-G6, $n$ = 6) (Fig. 3b). Although animals treated with Ads, OMVs, OMVs@P$_2$O, and OMVs-Ads had moderate tumor growth inhibition, the highest suppression of TC-1-hCD46 tumor growth was by the microbial nanocomposite (Fig. 3c–e, and Supplementary Fig. 12). As illustrated in Supplementary Figure 13, the weight of mice during the treatment period was unchanged, depicting the safety profile of the microbial nanocomposite. Additionally, the infiltration of CD8$^+$ T cells in the tumor of mice treated with the microbial nanocomposite was dramatically improved compared to the other groups. This validated their efficiency in vivo antitumor potentials (Supplementary Fig. 14). Moreover, as illustrated in Fig. 3-h, an enzyme-linked immunosorbent assay (ELISA) was used to detect a significant increase in the immune cytokine levels in serum within the microbial nanocomposite group. Transcriptomic analysis of the tumor xenografts was conducted to investigate the changes in gene expression after OMVs@P$_2$O-Ads treatments and assess the expression of immune-related genes. As shown in Figs. 3i, 1804 genes were upregulated, and 296 were down-regulated among all the 16776 genes with detected expressions. According to the logFC value, the genes were ranked, and gene set enrichment analysis (GSEA) was performed to differentially cluster expressed gene transcripts with gene ontology (GO) terms and the Kyoto Encyclopedia of Genes and Genomes Pathways. The up-regulated genes were significantly enriched in immune pathways (Fig. 3j), as depicted by "Activation of immune response" (Fig. 3k).

## Preparation and in vivo biodistribution evaluation of the bio-mineralized microbial nanocomposite

Compared to i.t. administration, the intravenous injection (i.v.) of Ads was more efficacious in treating deep tumors and minute tumor lesions. However, the targeted delivery of Ads is limited by innate immune responses, thereby decreasing the in vivo circulation time. Herein, the biomineralization technique was adopted to improve the biocompatibility of the microbial nanocomposite and achieve the systematic delivery of Ads. TEM, DLS, and energy spectrum analyses established the successful preparation of the biomineralized microbial nanocomposite (Fig. 4a, b and Supplementary Fig. 15). After forming the "core-shell" biomineralized microbial nanocomposite, we evaluated whether the CaP shells could efficiently prevent Ads from in vivo clearance, thereby promoting Ads accumulation in the tumor. The biomineralized/non-biomineralized microbial nanocomposite was injected intravenously within the TC-1-hCD46 xenograft tumor-bearing C57 female mice at an equivalent dose of 10$^7$ pfu for Ads. The biomineralized microbial nanocomposite was targeted and accumulated within the tumor tissues instead of the liver after 24 h of treatment (Fig. 4c). The OVs contents in multiple organs and tumors were determined after 24 h of treatment with PBS to quantify the biodistribution of the biomineralized microbial nanocomposite, the non-biomineralized microbial nanocomposite (i.v.), the biomineralized microbial nanocomposite (i.v.) and Ads (i.t.) using real-time quantitative polymerase chain reaction (RT-qPCR) (Fig. 4d). The Ads content in tumors treated with the biomineralized microbial nanocomposite was increased by approximately 7% of viral contents in the tumor treated with Ads (i.t.) than the non-biomineralized microbial nanocomposite group. This was consistent with the outcome of fluorescence analysis. The microbial nanocomposite (i.v.) Ads were primarily detected in the liver and spleen after 24 h. Thus, the non-biomineralized microbial nanocomposite would be opsonized during systemic circulation. It was eliminated by the mononuclear phagocytic system and sequestered in the spleen by the reticuloendothelial system.

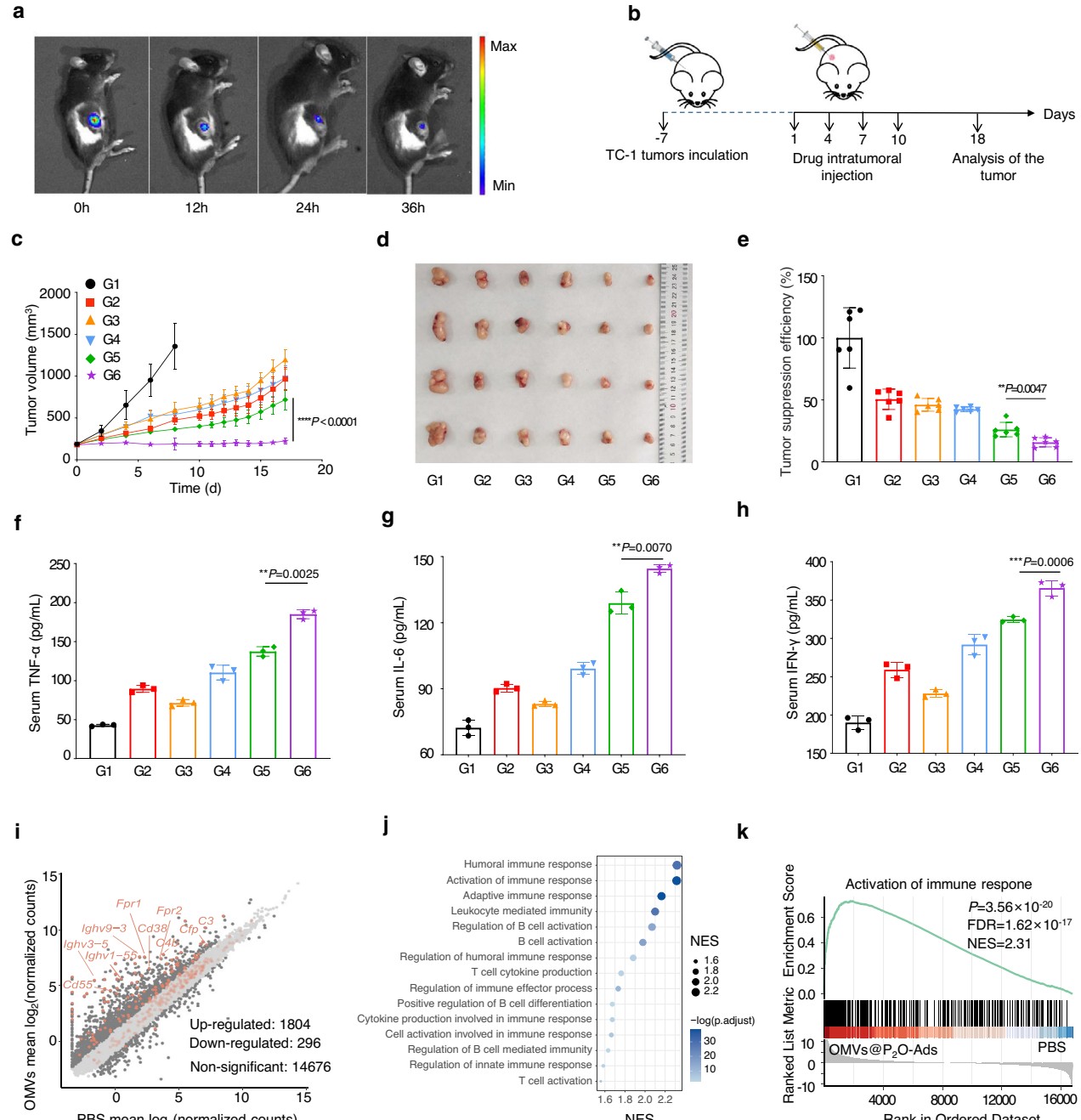

**Fig. 3 | In vivo oncolytic efficacy of the microbial nanocomposite. a** In vivo DIR fluorescent imaging of the microbial nanocomposite in TC-1-hCD46 xenograft tumor-bearing mice by IVIS ($n = 4$ mice). **b** Schematic illustration of the antitumor activity and immunology assessment experiments of the microbial nanocomposite using TC-1-hCD46 xenograft tumor-bearing C57 female mice model. TC-1 cells ($10^6$) were subcutaneously injected into the waist of female C57 mice, and the tumor-bearing mice were divided into six groups. **c** Tumor volume growth profiles of C57 mice bearing TC-1 xenografts. Data are presented as mean ± SD ($n = 6$ mice). ****$P < 0.0001$ by two-tailed Student's t-test. Source data are provided as a Source Data file. **d** Images of representative tumors of different treated groups on the 18th day ($n = 6$ mice). **e** Statistical graph of tumor weight of different treated groups on the 18th day. Data are presented as mean ± SD ($n = 6$ mice). **$P < 0.01$ by two-tailed Student's t-test. Source data are provided as a Source Data file. **f**–**h** Images of concentration of main cytokines in serum. Data are presented as mean ± SD ($n = 3$ independent experiments). **$P < 0.01$ and ***$P < 0.001$ by two-tailed Student's t-test. Source data are provided as a Source Data file. **i** The differential gene expression between the samples treated with OMVs@$P_2O$-Ads and PBS, using the absolute value of logFC greater than 1 as the threshold ($n = 2$ mice). j GSEA enriched pathways of the upregulated genes in the samples treated with OMVs@$P_2O$-Ads, showing immune-related terms. **k** Gene set enrichment analysis (GSEA) of the term "Activation of immune response", and the genes included in this pathway are highlighted in (i) with light yellow brown. $P$ values for GSEA were calculated using two-sided permutation test, and adjusted using Benjamini-Hochberg methods. (G1: PBS, G2: Ads, G3: OMVs, G4: OMVs@$P_2O$, G5: OMVs-Ads, G6: OMVs@$P_2O$-Ads).

The level of ROS expression, autophagy, and Ads replication was examined in the tumor after 72 h of different treatments. The tumors were separated from the mice after treatment using PBS, the biomineralized microbial nanocomposite without $P_2O$, and biomineralized microbial nanocomposite with $P_2O$ (at an Ads equivalent dose of $10^7$ pfu) through dissection. The tumors were stained using DHE dye (dihydroethidium, a ROS marker), LC3 B protein fluorescence-labeled antibodies (an autophagic marker), and p62 protein fluorescence-

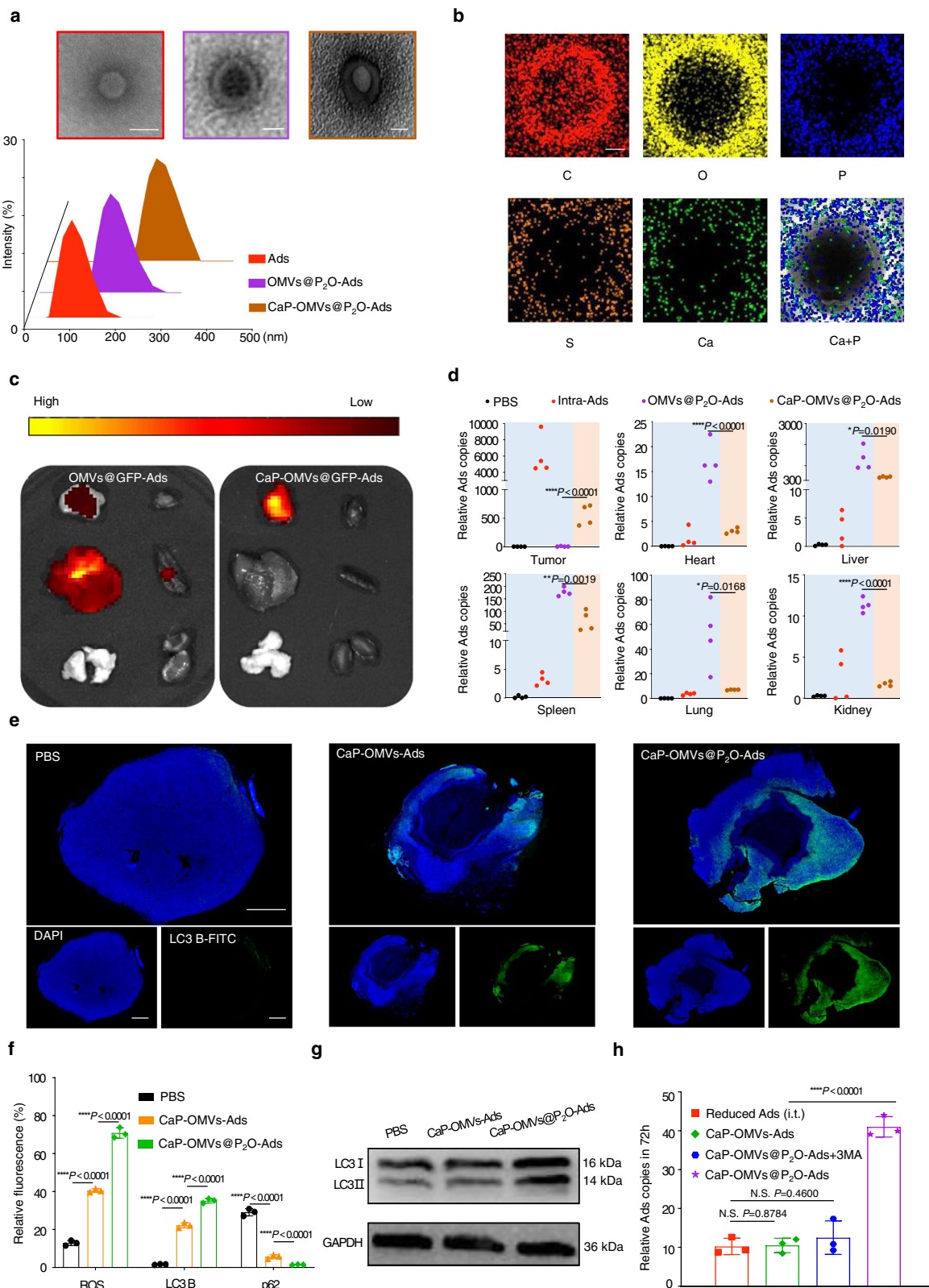

labeled antibodies (an autophagic marker). Supplementary Fig. 16 shows the strongest fluorescence of DHE in the biomineralized microbial nanocomposite group. Therefore, the $P_2O$ microbial nanocomposite could have a more potent capability to synthesize ROS. Besides, the autophagy induction ability of biomineralized microbial nanocomposite was verified depending on the strongest LC3 B protein fluorescence and the lowest p62 protein fluorescence (Fig. 4e and

Supplementary Fig. 17)[20]. Quantitative analyses of fluorescence intensity are shown in Fig. 4f. Additionally, the microbial nanocomposite treatment increased LC3-II/LC3-I ratio than without $P_2O$ (Fig. 4g and Supplementary Fig. 18). Thus, engineered $P_2O$-generated ROS could activate autophagy. Furthermore, the contents of Ads in these tumors were determined using RT-qPCR. Previously, it was observed that the Ads content in tumors was approximately 7% of administered Ads

**Fig. 4 | Preparation and in vivo evaluation of the biomineralized microbial nanocomposite. a** TEM and size distribution images of Ads, OMVs@P$_2$O-Ads, and CaP-OMVs@P$_2$O-Ads. Scale bar=100 nm. **b** Energy spectrum analysis image of the biomineralized composite microbe. Scale bar=50 nm. This experiments in (**a–c**) were repeated three times independently with similar results. **c** In vivo fluorescence imaging of the multiple organs and tumors collected from the mice at 24 h post i.v. injection. From left to right: tumor, heart, liver, spleen, lung, and kidney. **d** Quantitation of the biodistribution of relative Ads contents in multiple organs and tumors after 24 h of different treatments by RT-qPCR. Data are presented as mean ± SD ($n = 4$ mice). *$p < 0.05$, **$P < 0.01$ and ****$p < 0.0001$ by two-tailed Student's t-test. Source data are provided as a Source Data file. **e** Immunofluorescence images of LC3 autophagic proteins in tumor tissues. Blue represents DAPI-stained tumor cells and the green represents FITC-stained LC3 autophagic protein. Scale bar=2 mm. This experiment was repeated three times independently with similar results. **f** Quantitative analysis of fluorescence intensity. Data are presented as mean ± SD ($n = 3$ independent experiments). ****$p < 0.0001$ by two-tailed Student's t-test. Source data are provided as a Source Data file. **g** The expression of autophagy-related protein LC3-I and LC3-II examined by western blot. This experiment was repeated three times independently with similar results. **h** Quantitation of relative Ads content in the tumor after 72 h of different treatments by RT-qPCR technique. Data are presented as mean ± SD ($n = 3$ independent experiments). N.S. (No Significance) $P > 0.05$ and ****$p < 0.0001$ by two-tailed Student's t-test. Source data are provided as a Source Data file.

content after treatment with intravenous-injected biomineralized microbial nanocomposite. The reduced bare Ads at $7 \times 10^5$ pfu (7% of $10^7$ pfu) were intratumorally injected as the negative control. Figure 4H represents the Ads contents in groups treated with reduced Ads (i.t.), CaP-OMVs-Ads (i.v.), and CaP-OMVs@P$_2$O-Ads plus 3MA were statistically insignificant. The intratumoral content of Ads in the biomineralized microbial nanocomposite (i.v.) group was visibly increased. Therefore, the P$_2$O-contained biomineralized microbial nanocomposite could enhance the replication of Ads by inducing autophagy within tumor cells.

## In vivo oncolytic efficacy and immunoactivation capacity of the biomineralized microbial nanocomposite

The in vivo antitumor activity of the biomineralized microbial nanocomposite was assessed on the TC-1-hCD46 xenograft tumor-bearing C57 female mice model with an Ads equivalent dose of $10^7$ pfu (Fig. 5a). The OMVs@P$_2$O-Ads (i.v.) and CaP-OMVs-Ads (i.v.) groups moderately inhibited tumor growth. However, the biomineralized microbial nanocomposite (i.v.) possessed the highest capability in suppressing TC-1-hCD46 tumor growth. Bare Ads were administrated at $10^7$ pfu intratumorally as a positive control. Interestingly, the biomineralized microbial nanocomposite (i.v.) and Ads (i.t.) indicated a minimal difference in tumor inhibition (Fig. 5b, Supplementary Figs. 19 and 21). No significant changes were observed in the body weights of the animals and HE staining images of major organs. Additionally, hepatorenal functional indexes such as aspartate aminotransferase, alanine aminotransferase levels, blood urea nitrogen, and creatinine were also unchanged between the PBS-treated mice and the biomineralized microbial nanocomposite. Therefore, the biomineralized microbial nanocomposite showed a good in vivo safety profile (Supplementary Figs. 20, 22, and 23).

Furthermore, representative tumors were dissected after separating from the mice treated with six different agents. They were stained with CD8$^+$ fluorescence-labeled antibodies to assess the infiltration of CD8$^+$ T cells within the tumor tissues (Fig. 5c). Tumors, spleens, and blood were collected and disposed of to elucidate the mechanisms of antitumor activities of the biomineralized microbial nanocomposite and obtain single-cell suspensions for flow cytometry. TC-1-hCD46-bearing mice receiving treatment with biomineralized microbial nanocomposite revealed distinctly increased intratumoral infiltration of cytotoxic T lymphocyte (CD45$^+$CD3$^+$CD8$^+$ T cells) (Figs. 5d, g and Supplementary Fig. 39). CD8$^+$ T cell proportion in the tumors of animals treated with the biomineralized microbial nanocomposite (i.v.) was similar to those treated with Ads (i.t.). The enriched Ads in the tumor regions and replicating augmented Ads from the biomineralized microbial nanocomposite could enhance the CD8$^+$ T cell infiltration in tumor tissues. Therefore, it led to enhanced antitumor responses. Next, the proportion of IFNγ$^+$CD8$^+$ T cells in a tumor was investigated to accurately assess the strength of antitumor immunity (Supplementary Figs. 24–25 and 42). A more significant reduction of regulatory T cells (Treg cells: CD45$^+$CD3$^+$CD4$^+$ FOXP3$^+$ T cells, also called suppressor T cells) was observed within the group

treated with the biomineralized microbial nanocomposite (i.v.) compared to other groups (Figs. 5e, h and Supplementary Fig. 40). Thus, the biomineralized microbial nanocomposite treatment could effectively remold immunosuppressive conditions. The dendritic cells (DCs) can recognize the neoantigen of the tumor. Herein, the proportion of CD45$^+$CD11C$^+$MHC-II$^+$ DCs in the biomineralized microbial nanocomposite-treated group was higher than in other groups, thereby boosting the DCs recruitment and maturity in tumors (Fig. 5f, i and Supplementary Fig. 41). Besides, macrophage polarization at the tumor site was analyzed in each group of mice (Supplementary Figs. 26–29 and 42–43). In addition, the biomineralized microbial nanocomposite-treated group had a remarkable enhancement of effector memory T cells (TEM) in the spleen (Supplementary Figs. 30–31 and 44). These results demonstrated that the biomineralized microbial nanocomposite could have an effective strategy for strong and long-term immune responses.

Furthermore, multiple experiments in mice validated the ability of the biomineralized microbial nanocomposite to activate immunity at the tumor site. Based on the same experimental procedure and grouping in Fig. 5a, mice were dissected after four consecutive administrations, and CD8 + T cells in tumors were extracted using relevant kits. CD8$^+$ T cells isolated from each group were incubated in vitro using TC-1 cells at a ratio of 100:1. CD8$^+$ T cells extracted from the biomineralized microbial nanocomposite treatment group had the highest efficiency of killing tumor cells in vitro. Thus, the antitumor CD8$^+$ T cell proportion in the biomineralized microbial nanocomposite treatment group was higher than in other groups (Supplementary Fig. 32). Besides, the TC-1 tumor-bearing mice were divided into three groups ($n = 5$). The mice were injected intratumorally with PBS, CaP-OMVs@P$_2$O-Ads plus CD8$^+$ T cells antibody, and CaP-OMVs@P$_2$O-Ads when the tumor reached 100–150 mm$^3$. The drug was given every three days for four consecutive times on days 0, 3, 6, and 9. The mice were sacrificed and dissected on day 12 (Supplementary Fig. 33).

## In vivo tumor recurrence and metastasis suppression through the biomineralized microbial nanocomposite

All marketed OVs products are delivered by intratumoral or topical administration rather than intravenous administration. Due to the poor enrichment of OVs at metastatic or recurrent sites after intratumoral injection, intratumoral administration of OVs achieved poor outcomes for treating refractory cancers. We tested whether the biomineralized microbial nanocomposite effectively inhibited tumor recurrence and metastasis on the TC-1-hCD46 xenograft tumor-bearing C57 female mice model at an Ads equivalent dose of $10^7$ pfu (Fig. 6a). When the primary tumor volume reached 300 mm$^3$, tumors were resected. The 5% residual tumors were left to evaluate postsurgical residual primary tumor recurrence after the different treatments. As shown in Supplementary Fig. 34 and Fig. 6b, the microbial nanocomposite (i.v.), CaP-OMVs-Ads (i.v.), and decreased Ads (i.t.) groups could moderately inhibit tumor recurrence. However, the biomineralized microbial nanocomposite (i.v.) demonstrated optimal

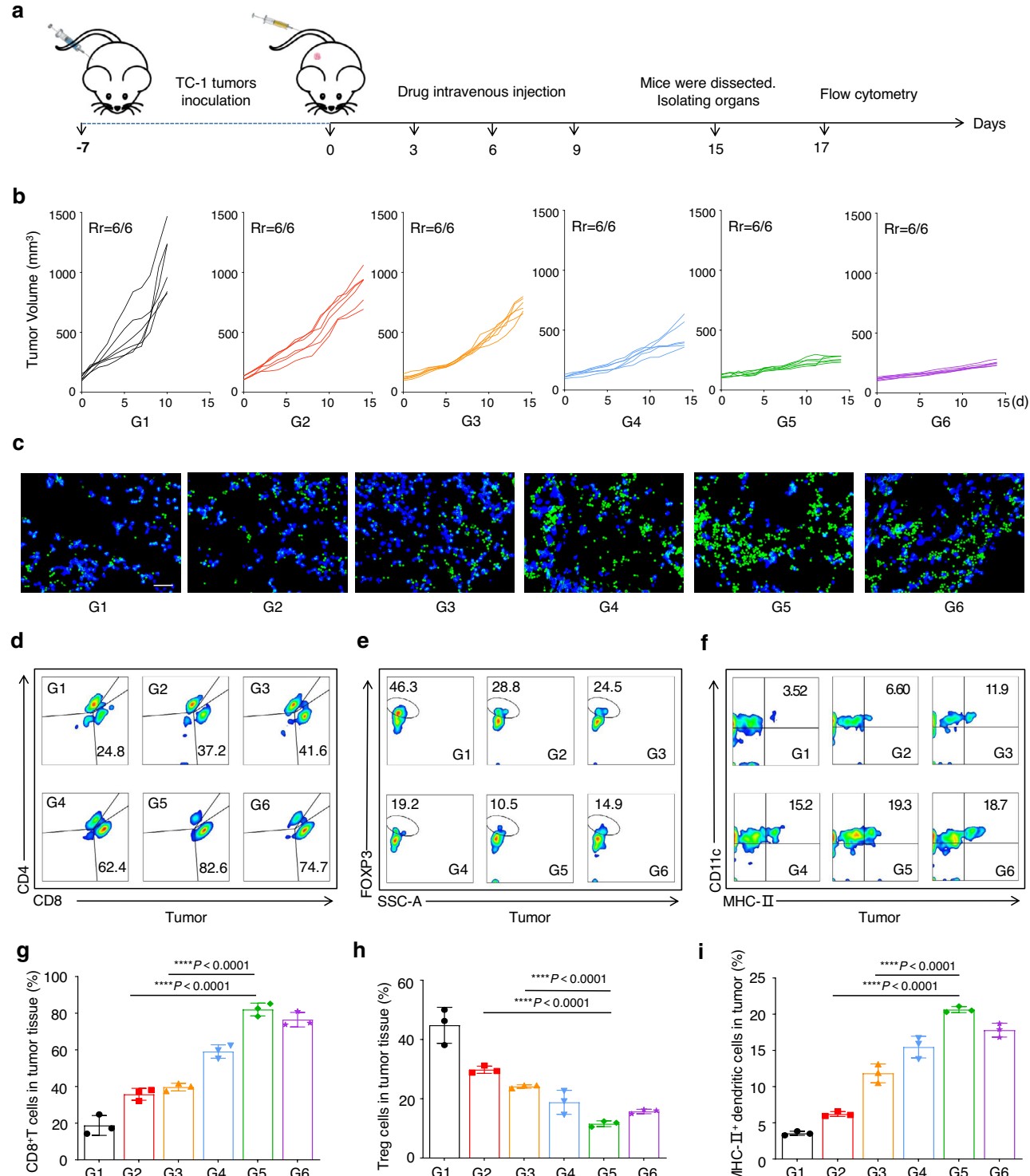

**Fig. 5 | In vivo oncolytic efficacy and immuneactivation capacity of the biomineralized microbial nanocomposite.** **a** Schematic illustration of the antitumor activity and immunity investigation of the biomineralized microbial nanocomposite on TC-1-hCD46 xenograft tumor-bearing C57 female mice model. **b** Individual tumor growth kinetics in different groups. Data are presented as mean ± SD (*n* = 6 mice). Source data are provided as a Source Data file. **c** The immunofluorescence images of CD8⁺ T cells in tumor tissues. Scale bars = 50 μm. This experiment was repeated three times independently with similar results. **d** Representative flow cytometric evolution images (**g**) as well as relative quantification of CD8⁺ T cells (CD45⁺CD3⁺CD8⁺) in the tumor. Data are presented as mean ± SD (*n* = 3 independent experiments). ****p < 0.0001 by two-tailed Student's t-test. Source data are

provided as a Source Data file. **e** Representative flow cytometric evolution images (**h**) as well as relative quantification of Treg cells (CD45⁺CD3⁺CD4⁺FOXP3⁺) in the tumor. Data are presented as mean ± SD (*n* = 3 independent experiments). ****p < 0.0001 by two-tailed Student's t-test. Source data are provided as a Source Data file. **f** Representative flow cytometric evolution images (**i**) and relative quantification of MHC-II⁺ DC cells (CD45⁺CD11C⁺MHC-II⁺) in the tumor. Data are presented as mean ± SD (*n* = 3 independent experiments). ****p < 0.0001 by two-tailed Student's t-test. Source data are provided as a Source Data file. (G1: PBS, G2: OMVs@P₂O-Ads, G3: CaP-OMVs-Ads, G4: Intra-Ads, G5:CaP-OMVs@P₂O-Ads, G6: Intra-Ads high does).

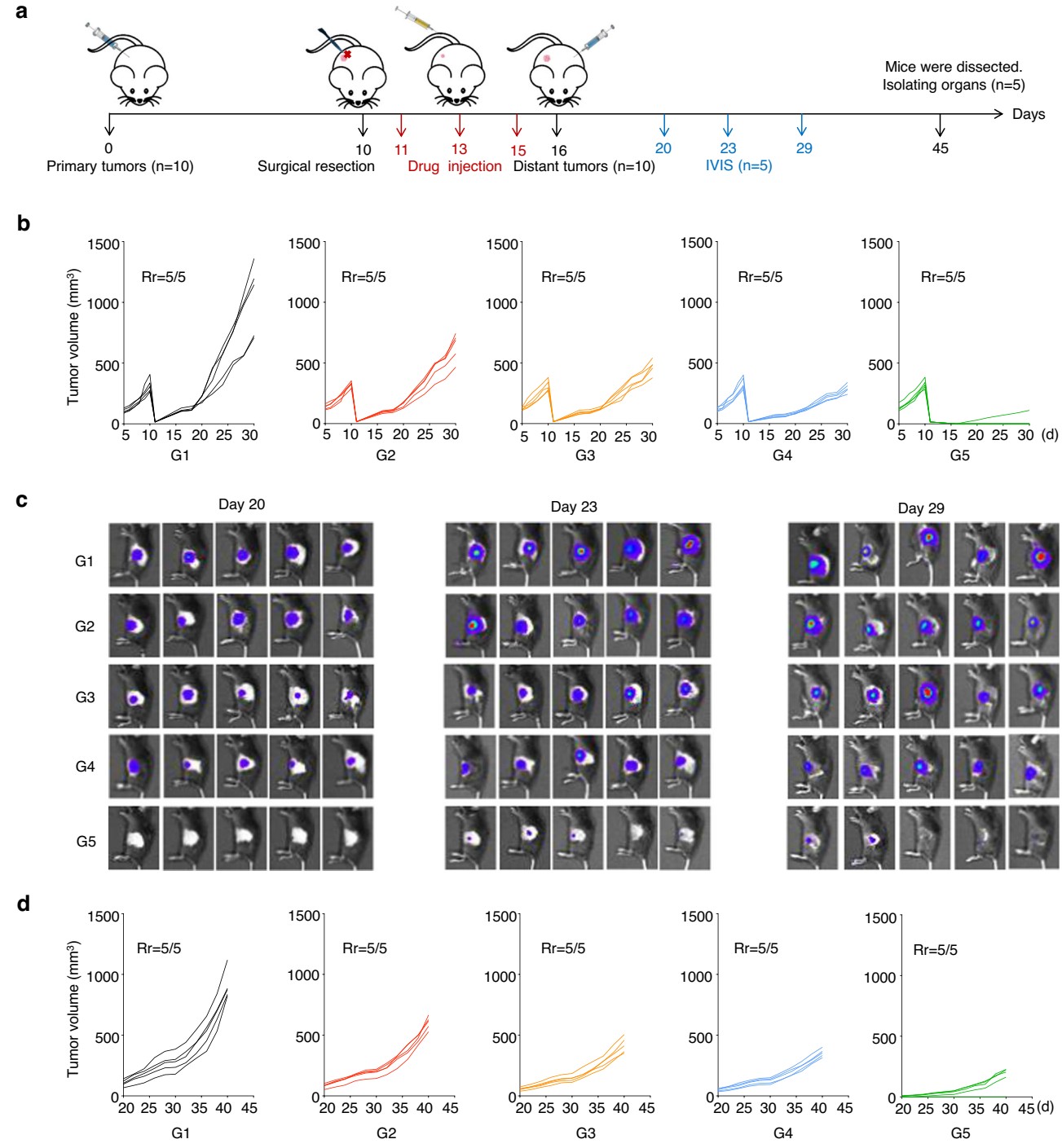

**Fig. 6 | The capacity of the biomineralized microbial nanocomposite for inhibiting tumor recurrence and metastasis in vivo. a** Schematic illustration of the inhibiting tumor recurrence and metastasis investigation of the biomineralized microbial nanocomposite on TC-1-hCD46 xenograft tumor-bearing C57 female mice model. **b** Individual primary tumor growth kinetics in different groups. Data are presented as mean ± SD ($n = 5$ mice). Source data are provided as a Source Data file. **c** In vivo bioluminescence imaging of distant tumors metastasis. ($n = 5$ mice). **d** Individual distant tumor metastasis kinetics in different groups. Data are presented as mean ± SD ($n = 5$ mice). Source data are provided as a Source Data file. (G1: PBS, G2: OMVs@P$_2$O-Ads, G3: CaP-OMVs-Ads, G4: Intra-Ads, G5:CaP-OMVs@P$_2$O-Ads).

capability in suppressing TC-1-hCD46 tumor recurrence. Among five mice, only one in the experimental group showed weak tumor recurrence, with a recurrence inhibition rate of 80% (Supplementary Fig. 36). Subsequently, the distal tumors were inoculated after 16 days of treatment to mimic tumor metastasis. IVIS and vernier calipers were used to measure distant tumor volumes. This established that the biomineralized microbial nanocomposite had the strongest ability

against tumor metastasis (Fig. 6c, d and Supplementary Fig. 37). Moreover, the mice receiving the biomineralized microbial nanocomposite treatment showed an extended survival period of >60 days and no significant change in body weight during the 45 days (Supplementary Figs. 35 and 38). Thus, the biomineralized microbial nanocomposite strategy has high therapeutic efficiency and good bio-safety.

## Discussion

Oncolytic virotherapy is an emerging immunotherapy inducing antitumor responses through selective self-replication inside cancer cells and oncolytic virus (OV)-mediated immunostimulation. It has attracted more attention recently. However, although OVT has incredible advantages in cancer treatment, the clinical practice of commercial OVs is not perfect. The three oncolytic viral drugs marketed globally are administered by intratumoral injection. This significantly increases the difficulty of clinical treatment and decreases medication compliance in patients. In addition, some clinical trials have attempted to deliver OVs systematically, with unsatisfactory clinical results.

We constructed the microbial nanocomposite for autophagy-cascade-augmented immunotherapy. The oncolytic Ads were encapsulated using the engineered OMVs extracted from *E. coli* and transfected with plasmid to express $P_2O$. CaP biomineral shells were added to protect Ads from the clearance of the innate immune system. Therefore, it extends the in vivo circulation time and promotes Ads enrichment after systemic administration. More importantly, $P_2O$-catalyzed $H_2O_2$ elevated the level of oxidative stress in the tumor site, leading to autophagy formation. The increase in the number of autophagy-induced autophagosomes would significantly augment the replication efficiency of Ads in OVs-infected cancer cells. Meanwhile, enhanced OVs intratumoral enrichment augmented OVs replication in tumors and immunosuppressive TEM remolding based on the advantage of the immunostimulatory capability of OMVs. This would enhance OVs-mediated immune responses. Overall, the current autophagy-cascade-boosted immunotherapy strategy would be promising in OVs-based biomedical therapy applications.

## Methods

### Ethical statement

Our research complies with all relevant ethical regulations. All the animal protocols were performed in line with the Guidelines for the Care and Use of Laboratory Animals and approved by the Institutional Animal Ethical Care Committee (IAEC) of Shenyang Pharmaceutical University.

### Cell lines and animals

HCT116 cell line, the modified non-small cell lung (TC-1) cancer cell line expressing the human CD46 receptor and TC-1-hCD46-luc cell line were the kindly gift from the Sino-British Research Centre of Zhengzhou University. Cell lines validation using short tandem repeat (STR) markers were performed by Genetic Testing Biotechnology Corporation (Suzhou, China). In detail, eighteen STR loci were amplified using multiplex PCR. One additional marker (Human TH01) was used to screen for the presence of human species. The cell line sample was processed using the ABI Prism 3130 XL Genetic Analyzer. Data were analyzed using Gene Mapper ID 3.2 software (Applied Biosystems). Appropriate positive and negative controls were run and confirmed for each sample submitted. All cell lines tested negative for mycoplasma contamination. The HCT116 cell line, TC-1-hCD46 cell line and TC-1-hCD46-luc cell line were maintained in Dulbecco's modified Egle's medium (DMEM) with 10% FCS.

C57BL/6JOlaHsd mice (female, 8–10 weeks old) were supplied by the Animal Center of Shenyang Pharmaceutical University (Shenyang, Liaoning, China). There were no restrictions on the sex of the experimental animals in this study. The living environment of animals were maintained at a temperature of ~25 °C with a 12 h light/dark cycle, with free access to standard food and water. The humane endpoints included tumor burden exceeding 10% of normal body weight, animal weight loss exceeds 20% of normal animal weight, ulcer at tumor growth point, and sustained self-mutilation in animals. The humane end-point was approved by Certification and Accreditation Administration of the People's Republic of China (CNCA). Cervical dislocation under deep anesthesia was adopted for euthanasia. All animal procedures were carried out under the guidelines approved by the Institutional Animal Care and Use Committee of Shenyang Pharmaceutical University.

### Software and code

The software used for data collection in this study includes Malvern: Zetasizer Software 7.01, Microplate reader: SkanIt 2.4.3.37 and Confocal laser scanning microscopy: NIS 4.13. The software used for data analysis in this study includes GraphPad prism 8.0, Microsoft Office 2019, ImageJ 1.8.0, FlowJo_V10, Trim-galore (version 0.6.7); HISAT2 (version 2.2.1), featureCounts (version 2.0.1), DESeq2 (version 1.32.0) and R package clusterProfiler, ABI Prism 3130 XL Genetic Analyzer, Gene Mapper ID 3.2 software (Applied Biosystems).

### Materials

The oncolytic adenovirus11-Tel (Ads) expressing GFP protein using telomerase (TERT) promoter and Ad5 enhancer was a gift from Beijing Bio-Targeting Therapeutics Technology Co., Ltd, and Zhengzhou University. *E. Coli*@$P_2O$, an engineered non-pathogenic bacterium, was provided by Shenyang Pharmaceutical University. The primers are synthesized by BGI Genomics Co.,Ltd. (No: BJP12162201823; Primer ID: 221216_020E07; 221216_020F07). Sequence (5′ to 3′): TTGGACGGCTCCTGGAATAG; Sequence (5′ to 3′): TGTGACGGAAACAACCCTGACT. Details of the antibodies used in this study are presented in Supplementary Table 1.

### Preparation and characterization of OMVs@$P_2O$-Ads

The pyranose oxidase ($P_2O$) plasmid was expressed on engineered *E. coli* through transfection. Subsequently, the $P_2O$-expressing bacteria (*E. coli*@$P_2O$) were cultured overnight in a traditional Luria-Bertani medium at 37 °C. Then, 0.5 mM isopropyl-β-D-thiogalactoside was added at 15 °C, shaken for 20 h, and centrifuged 10 min at $900 \times g$ at 4 °C. The supernatant was discarded, and collected bacteria were resuspended in 1 mL PBS followed by a 20 times dilution, addition of 200 μL 5 mg/mL glucose solution, and incubation at 37 °C for 1 h. Then, the sample interacted with 200 μL of 1 mg/mL ABTS (2, 2-azo-bis (3-ethyl-benzothiazole-6-sulfonic acid) diammonium salt) and 200 μL of 1 mg/mL horseradish peroxidase for 12 h at 4 °C. This could prove the presence of pyran oxidase if the sample turned blue. Next, the bacterial liquid was transferred into centrifugal tubes and centrifuged at $1050 \times g$ for 0.5 h. The precipitate was discarded, and the supernatant was collected and filtered through a 0.45 μm aseptic filter membrane. The filtrate was concentrated 100 times using a 100,000 MWCO ultrafiltration concentrator and centrifuged at high speed ($200,000 \times g$) for 2 h. The precipitate was collected, resuspended with a one-tenth concentration of PBS, filtered through a 0.45 μm membrane, and then stored at 4 °C. The sample was filtered again through a 0.45 μm membrane before use. The analogous method confirmed pyranose oxidase in bacterial outer membrane vesicles (OMVs@$P_2O$). For OMVs@$P_2O$-Ads preparation, OMVs@$P_2O$ (1 mg total protein) and Ads (0.5 mg total protein) were dispersed in 1 mL of DMEM medium. The dispersion was extruded 20 times through a 400 μm filter membrane using a liposome extruder. Then, the extruded OMVs@$P_2O$-Ads dispersion was collected and stored at 4 °C. Particle size and surface potential were determined using Malvern (Zetasizer Software 7.01).

### Preparation and characterization of CaP-OMVs@$P_2O$-Ads

For CaP-OMV@$P_2O$-Ads preparation, OMVs@$P_2O$-Ads (1 mg total protein) were dispersed in 1 mL of DMEM medium by mixing overnight in a mixer at 4 °C to ensure equilibrium. Then, 10 μL of $CaCl_2$ (1 M) was added to the reaction mixture, and the incubation was continued at 37 °C for another 2 h. Next, the biomineralized CaP-OMV@$P_2O$-Ads were washed thrice with ultrapure water by centrifugation at $20,000 \times g$ for 20 min. The as-obtained sediment was resuspended

using PBS for further characterization. The yield was determined by comparing the changes in protein concentration of bionic mineralization. Particle size and surface potential were determined using Malvern: Zetasizer Software 7.01.

## Producing ROS in vitro assay

TC-1 mouse lung cancer cells were cultured in DMEM medium supplemented with 10% FBS, penicillin (100 U mL$^{-1}$), and streptomycin (100 µg mL$^{-1}$). The cells were maintained in a humidified atmosphere of 5% $CO_2$ at 37 °C. TC-1 cells were cultured and divided into three groups (PBS, OMVs, and OMVs@$P_2$O). After the cells occupied 80% of the bottom, the medium was discarded, and the cells were rinsed twice using PBS. DCFH-DA fluorescent dye (10 µM, 1 mL) (Meilun ROS Assay Kit MA0219) was added to the blank medium working solution. The cells were then incubated at 37 °C for 1 h in the dark. Next, the medium was discarded, and the cells were rinsed with PBS again. Then, PBS, OMVs, and OMVs@$P_2$O blank medium dispersion were added sequentially and further incubated at 37 °C for 3 h in the dark. After rinsing with PBS, the cells were collected, and their ROS concentration was determined by flow cytometry. The extracellular DCFH-DA has no fluorescence even after possessing the capability of crossing the cell membrane freely. After entering the cell, it can be hydrolyzed by intracellular esterase to translate into DCFH, which cannot pass through the cell membrane. In the presence of ROS, DCFH is oxidized to produce the fluorescent substance DCF (the excitation wavelength: 502 nm; the emission wavelength: 530 nm).

## Evaluation of autophagosome in vitro

When cells covered 80% of the bottom, the medium was discarded and washed twice with PBS. PBS, OMVs-Ads, and OMVs@$P_2$O-Ads blank medium dispersion were added sequentially and incubated at 37 °C for 3 h. Next, dansylcadaverine dye and Ethidium bromide dye were added successively. The fluorescence intensity of each group was then observed under CLSM. Related data was collected using Confocal laser scanning microscopy (NIS 4.13).

## Evaluation of LC3 protein in vitro (Western blot)

$5 \times 10^6$ cells were cultured and incubated with indicated PBS, OMVs-Ads, and OMVs@$P_2$O-Ads. After incubation for 24 h, RIPA buffer was added to destroy the cells. The supernatants were harvested and clarified using centrifugation at $12,000 \times g$ for 3 min. Loading buffer diluent was added to make the total protein content consistent across the three groups. The samples were boiled in the water bath for 10 min and cooled for storage. The proteins were dissociated by 15% SDS-PAGE. Then, the gel was transferred onto PVDF membranes and blocked by 5% bovine serum albumin (BSA) for 1 h. Subsequently, LC3B protein rabbit pAb (1:500 A5601, ABclonal) was employed at 4 °C overnight. The membranes were incubated with appropriate secondary antibodies (1:500; AS011, ABclonal) for another 1 h the next day. All the membranes were rinsed in three cycles of 15 min each using tris-buffered saline with tween (TBST). Finally, the target bands were exposed and visualized with the ECL detection system (Bio-Rad, USA). Besides, internal reference was set by GAPDH.

## In vitro evaluation of the effect of microbial nanocomposite on Ads replication

TC-1 cells were cultured in cell culture plates. After the cells occupied 80% of the bottom, the medium was sucked out with a 2 mL syringe and washed with PBS twice. Ads, OMVs-Ads, OMVs@$P_2$O-Ads plus 3MA, and OMVs@$P_2$O-Ads were added to each group and incubated for 3 h at 37 °C. 3MA is an autophagy inhibitor: 3-Methyladenine. The drug solution was dumped out, the cells were rinsed twice with PBS, and the same volume of blank medium was added. Plates were put in 37 °C and moved after 0, 24 h, 36 h, and 48 h, and were placed at −80 °C for three times to ensure complete cell breakdown. The freeze-thaw solution was collected and centrifuged at $840 \times g$ at 4 °C for 0.5 h. Precipitation was discarded, and 1% triton was added to the supernatant. Then, the liquid was pre-denatured at 98 °C, and the precipitate of deformed protein was removed by centrifugation at $3600 \times g$ for 10 min. Finally, the RT-qPCR technique collected and processed the supernatant for quantitative Ads detection. All the reagents of the RT-qPCR technique were purchased from Vazyme.

## Cytotoxicity assay

TC-1 cells were cultured in cell culture plates. After the cells occupied 80% of the bottom, the medium was sucked out with a 2 mL syringe and washed twice with PBS. Then, PBS, Ads, OMVs, OMVs@$P_2$O, OMVs-Ads, and OMVs@$P_2$O-Ads were added and incubated at 37 °C for 24 h. Next, the cells were separately prepared for fluorescence analysis and MTT assay[21]. To determine fluorescence intensity, calcein-AM, and propidium iodide (PI) dyes were added, and CLSM observed each group of cells. On the other hand, for the MTT assay, 150 µL of MTT (1 mg/mL) reagent was introduced into the cells, which were incubated for another 24 h at 37 °C. After removing the MTT solution, 100 µL of dimethyl sulfoxide was added, and the absorbance was measured at 570 nm on a microplate spectrophotometer using Microplate reader (SkanIt 2.4.3.37). The method of HCT116 cytotoxicity assay is the same as above.

## Evaluation of the intratumoral accumulation of OMVs@$P_2$O-Ads after intratumoral injection

C57 mice (female, 8–10 weeks old) were obtained from the Laboratory Animal Center of Shenyang Pharmaceutical University. The animal experiments were performed by following the Guidelines for the Care and Use of Laboratory Animals approved by the IAEC of Shenyang Pharmaceutical University.

OMVs@$P_2$O-Ads was prepared as described in "Preparation and characterization of OMVs@$P_2$O-Ads", and an excess of DIR staining solution was subsequently added to label OMVs. The free DIR dye was removed by centrifuging at $3000 \times g$ for 3 min using an ultrafiltration tube with a 100 kDA pore size. DIR-labeled OMVs@$P_2$O-Ads were injected intratumorally using Ads content of $7 \times 10^5$ PFU as a standard. DIR fluorescent imaging of the microbial nanocomposite in vivo in TC-1-hCD46 xenograft tumor-bearing mice by IVIS.

DIR is a type of long-chain lipophilic dialkylcarbocyanine dye. Owing to its lipophilicity, DIR is often used to label cell membranes as well as other liposoluble biological structures including OMVs. The maximum excitation and emission wavelengths of DIR are 750 nm and 780 nm, respectively. Because the infrared light emitted by DiR can efficiently pass through cells and tissues, it is of great significance in in vivo imaging or tracking.

## Evaluation of the antitumor effect of microbial nanocomposite in vivo

C57 mice (female, 8–10 weeks old) were obtained from the Laboratory Animal Center of Shenyang Pharmaceutical University. The animal experiments were carried out based on the Guidelines for the Care and Use of Laboratory Animals approved by the IAEC of Shenyang Pharmaceutical University.

TC-1 cells ($10^6$) were subcutaneously injected into the waist of female C57 mice, and the tumor-bearing mice were divided into six groups ($n = 6$). When the tumor reached 100–150 mm$^3$, the mice were injected intratumorally with PBS, Ads ($7 \times 10^5$ PFU), OMVs, OMVs@$P_2$O, OMVs-Ads, and OMVs@$P_2$O-Ads ($7 \times 10^5$ PFU). The drug was given every 3 days for four consecutive times, the tumor volume was measured with a vernier caliper, and mice were weighed daily. On the 18th day of the efficacy experiment, the mice were sacrificed by cervical spine removal, and the tumor tissue was isolated, weighed, and photographed. The effects of different preparations on tumor growth were analyzed and compared.

## Flow cytometry analysis

Peripheral blood, spleens, and tumor tissues were extracted from TC-1-hCD46-bearing C57 mice (female, 8–10 weeks old). Spleen and tumor tissue were first used to create a single-cell suspension. The cells were then stained with fluorescence-labeled antibodies. The antibody information is shown in Supplementary Table 1. Flow cytometry was used to measure the proportion of stained cells in the tumor tissue, and data were analyzed using FlowJo software.

## RNA-seq library construction and data processing

Cell-derived xenografts treated with OMVs@P$_2$O-Ads or PBS were collected, and then the total RNA was isolated using RNAiso Plus (TaKaRa). After removal of the remaining genomic DNA, mRNA was purified from total RNA using polyT and then fragmented with 10× RNA fragmentation buffer (100 mM Tris-HCl, 100 mM ZnCl$_2$ in nuclease-free H$_2$O). The RNA-seq library was constructed using Hieff NGS® Ultima Dual-mode mRNA Library Prep Kit for Illumina (Yeasen) according to the manufacturer's protocol and was sequenced using Illumina NovaSeq 6000.

Raw reads were trimmed by Trim-galore (version 0.6.7) with the parameter '--paired' and were aligned to the mouse genome (GRCm39/mm39) using HISAT2 (version 2.2.1)[22] with default parameters. Unmapped reads and non-uniquely mapped reads were filtered out by samtools[23]. Counts for annotated genes were generated using featureCounts (version 2.0.1)[24] with the GENCODE VM30 annotation file[25]. The raw counts were normalized using the estimateSizeFactors function of the DESeq2 (version 1.32.0) to identify differentially expressed genes in RNA-seq[26], The P and logFC values were calculated using the DESeq function. R package clusterProfiler[27] was used to perform GSEA analyses.

## In vivo biodistribution qualitative investigation

TC-1 cells (10$^6$) were subcutaneously injected into the waist of C57 mice (female, 8–10 weeks old). And tumor-bearing mice were divided into two groups. One group received intravenous OMVs@GFP-Ads, whereas the other group received intravenous CaP-OMVs@GFP-Ads. After 24 h of treatment, the mice were sacrificed by cervical spine removal, and their organs were isolated for ex vivo fluorescence imaging.

## In vivo biodistribution quantitative investigation

TC-1 cells (10$^6$) were subcutaneously injected into the waist of C57 mice (female, 8–10 weeks old). And tumor-bearing mice were divided into four groups. PBS, OMVs@P$_2$O-Ads, and CaP-OMVs@P$_2$O-Ads were intravenously injected into three separate groups. In contrast, Ads were injected intratumorally into the fourth group. The mice were sacrificed by cervical spine removal after 24 h of the treatment. DNA was extracted from the tumor tissues following their dissection from the mice. RT-qPCR was used to detect the content of Ads in the tumor.

## Producing ROS in vivo assay

Circle and Spontaneous fluorescence quenching: frozen slides were restored to room temperature. The clear liquid was eliminated, and the objective tissue was marked with a liquid blocker pen. Then spontaneous fluorescence quenching reagent was added to incubate for 5 min, washing in running tap water for 10 min. Staining: ROS staining solution was added to the marked area, incubated at 37 °C for 30 min, and kept in the dark place. DAPI counterstain in the nucleus: washing frozen section three times with PBS (pH 7.4) in a Rocker device for 5 min each. Then, the frozen section was incubated with DAPI solution at room temperature for 10 min and kept in the dark place. Mount: frozen section was washed thrice with PBS (pH 7.4) in a Rocker device for 5 min each. The liquid was thrown away slightly and then covered by an anti-fade mounting medium. Images were detected by Microscopy and collected by Fluorescent Microscopy.

## Evaluation of LC3 protein in vivo (Western blot)

The tumor tissues from each mice group were broken into pieces of 200–250 mm$^3$ using the homogenizer. Then, RIPA buffer was added to destroy the cells and incubated for 4 h. Supernatants were harvested and clarified by centrifugation at 12,000 × $g$ for 3 min. The loading buffer diluent was added to make the total protein content consistent in the three groups. The samples were boiled in the water bath for 10 min and stored at −80 °C. The proteins were dissociated by 15% SDS-PAGE. The gel was transferred onto PVDF membranes and blocked by 5% BSA for 1 h. Subsequently, LC3B protein rabbit pAb (1:500 A5601, ABclonal) was employed at 4 °C overnight. The following day, membranes were incubated with appropriate secondary antibodies (1:500; AS011, ABclonal) for another 1 h. All the membranes were rinsed using TBST for 15 min three times. Finally, the target bands were exposed and visualized using the ECL detection system (Bio-Rad, USA). Besides, the internal reference was set by GAPDH.

## Autophagy in vivo assay (Immunofluorescence)

Deparaffinize and rehydrate: paraffin section was incubated sections in two changes of xylene for 15 min each, rehydrated in two changes of pure ethanol for 5 min, and followed by dehydrating in gradient ethanol of 85% and 75% ethanol, respectively, for 5 min each. Then, the paraffin section was washed in distilled water. Antigen retrieval: the slides were immersed in EDTA antigen retrieval buffer (pH 8.0), remaining at a sub-boiling temperature for 8 min, standing for 8 min, and then at another sub-boiling temperature for 7 min. It is essential to prevent the evaporation of the buffer solution. After cooling, the paraffin section was washed thrice with PBS (pH 7.4) in a Rocker device for 5 min each. The right antigen retrieval buffer and heat extent were used according to tissue characteristics. Circle and serum blocking: clear liquid was eliminated, and the objective tissue was marked with a liquid blocker pen. Next, 3% BSA was added to cover the marked tissue to block non-specific binding for 30 min. The objective area was covered with 10% donkey serum (for the case of primary antibody originating from goat) or 3% BSA (for the case of primary antibody arising from others). Primary antibody: the blocking solution was thrown away slightly. Slides were incubated with primary antibody (diluted with PBS appropriately) overnight at 4 °C and placed in a wet box containing a little water. Secondary antibody: slides were washed thrice with PBS (pH 7.4) in a Rocker device for 5 min each. Then the liquid was thrown away slightly. Objective tissue was covered with secondary antibody (appropriately respond to primary antibody in species) and incubated at room temperature for 50 min in dark conditions. DAPI counterstain in the nucleus: paraffin section was washed three times with PBS (pH 7.4) in a Rocker device, 5 min each, then incubated with DAPI solution at room temperature for 10 min and kept in a dark place. Spontaneous fluorescence quenching: The paraffin section was washed thrice with PBS (pH 7.4) in a Rocker device for 5 min each. Spontaneous fluorescence quenching reagent was added to incubate for 5 min, washing in tap water for 10 min. Mount: liquid was thrown away slightly and then was covered by a coverslip with an anti-fade mounting medium. Images were detected by Microscopy and collected by Fluorescent Microscopy. DAPI emits blue light with a UV excitation wavelength of 330–380 nm and emission wavelength of 420 nm; FITC emits green light with an excitation wavelength of 465–495 nm and emission wavelength of 515–555 nm.

## In vivo evaluation of the effect of microbial nanocomposite on Ads replication

TC-1 cells (10$^6$) were subcutaneously injected into the waist subcutaneous of female C57 mice. When the tumor reached the size of 200–250 mm$^3$, the mice were divided into four groups ($n = 3$). Ads were administered intratumorally in one group, while the CaP-OMVs-Ads CaP-OMVs@P$_2$O-Ads plus 3MA and CaP-OMVs@P$_2$O-Ads were injected intravenously in the other groups. The tumor tissue was

dissected from the mice after 72 h, and DNA was extracted from the tumor tissue. RT-qPCR was used to detect the content of Ads in the tumor.

### Evaluation of the antitumor effect of CaP-OMVs@P$_2$O-Ads in vivo

TC-1 cells ($10^6$) were subcutaneously injected into the waist of C57 mice (female, 8-10 weeks old), and tumor-bearing mice were divided into six groups ($n = 6$). When the tumor reached 100-150 mm$^3$, the mice were intravenously injected with PBS, OMVs@P$_2$O-Ads, CaP-OMVs-Ads, and CaP-OMVs@P$_2$O-Ads, while the Ads ($7 \times 10^5$ and $10^7$) were injected intratumorally. The drug was given every two days for four consecutive times. The tumor volume was measured using a vernier caliper, and mice were weighed daily. On the 14th day of the efficacy experiment, the mice were sacrificed by cervical spine removal, and the tumor tissue was isolated, weighed, and photographed. The effects of different preparations on tumor growth were analyzed and compared. Immunology was studied in the same way.

### Flow cytometry analysis

Peripheral blood, spleens, and tumor tissues were extracted from the TC-1-hCD46-bearing C57 mice (female, 8-10 weeks old). Spleen and tumor tissue were first used to create a single-cell suspension. The antibody information is shown in Supplementary Table 1. Flow cytometry measured the proportion of stained cells in the tumor tissue, and data were analyzed using the FlowJo software.

### Co-culture assay in vitro to verify the tumor-killing activity of CD8$^+$ T cells

First, TC-1 cells ($10^6$) were subcutaneously injected into the waist of C57 mice (female, 8-10 weeks old), and tumor-bearing mice were divided into six groups ($n = 6$). When the tumor reached 100-150 mm$^3$, the mice were intravenously injected with PBS, OMVs@P$_2$O-Ads, CaP-OMVs-Ads, and CaP-OMVs@P$_2$O-Ads, while the Ads ($7 \times 10^5$ and $10^7$) were injected intratumorally. CD8$^+$ T cells were extracted from each administration group based on the instructions of the BeaverBeads™ mouse CD8$^+$ T cell sorting kit (purchased from Beaver, 70903-100). Then, TC-1 cells were cultured in cell culture plates. After the cells occupied 80% of the bottom, the medium was sucked out with a 2 mL syringe and washed twice with PBS. Then, CD8$^+$ T cells extracted from each administration group were added into the holes based on the proportion of TC-1 cells: CD8$^+$ T cells = 1:100 and subsequently incubated at 37 °C for 24 h. Next, the MTT assay helped investigate the tumor-killing rate in each group.

### Evaluation of dependency of CD8$^+$ T cells

TC-1 cells ($10^6$) were subcutaneously injected into the waist of C57 mice (female, 8-10 weeks old), and tumor-bearing mice were divided into three groups ($n = 5$). When the tumor reached the size of 100-150 mm$^3$, the mice were intravenously injected with PBS, CaP-OMVs@P$_2$O-Ads, and CaP-OMVs@P$_2$O-Ads plus CD8$^+$ T cells antibody (anti-CD8 antibodies, clone: 2.43, Bio X cell, cat. no.: BP0061, injected i.v. every 2 days starting one day before the CaP-OMVs@P$_2$O-Ads injection) at 0, 3, 6, 9 day. On the 12th day of the efficacy experiment, the mice were sacrificed by cervical spine removal, and the tumor tissue was isolated, weighed, and photographed. The effects of the different preparations on tumor growth were analyzed and compared.

### Evaluation of inhibiting tumor recurrence and metastasis effect of CaP-OMVs@P$_2$O-Ads in vivo

TC-1 cells ($10^6$) were subcutaneously injected into the waist of C57 mice (female, 8-10 weeks old), and tumor-bearing mice were divided into five groups ($n = 5$). When the tumor reached the size of 300-500 mm$^3$, 95% of the primary tumor was removed via surgery. Then the mice

were intravenously injected using PBS, OMVs@P$_2$O-Ads, CaP-OMVs-Ads, and CaP-OMVs@P$_2$O-Ads, while the Ads ($7 \times 10^5$ pfu) were injected intratumorally. The drug was given every two days for three consecutive times, the tumor volume was measured with a vernier caliper, and mice were weighed daily. The mice were inoculated with distant tumors on the 16th day of the efficacy experiment. IVIS and vernier caliper was used to measure the volume of the tumor timely. On the 45th day of the efficacy experiment, the mice were sacrificed by cervical spine removal, and the tumor tissue was isolated, weighed, and photographed. The effects of different preparations on tumor growth were analyzed and compared.

### Safety

Body weights were monitored during the above pharmacodynamics. After the pharmacodynamics, the whole blood was collected for blood analysis. The serum was obtained for hepatorenal function analysis. The main organs (heart, liver, spleen, lung, and kidney) and tumor tissues were collected for hematoxylin and eosin (H&E) staining.

### Reporting summary

Further information on research design is available in the Nature Portfolio Reporting Summary linked to this article.

## Data availability

The next-generation-sequencing data generated by this study have been deposited to GEO database under accession number: GSE225094. The source data underlying Figs. 2a, f, h, 3c, e, f–h, 4a, d, f–h, 5b, g–i, 6b, d, Supplementary Figs. 4, 5, 8–10, 12–15, 18–20, 22, 25, 27, 29, 31–33, 35 and 38 are provided with this paper and are also available in figshare at: https://doi.org/10.6084/m9.figshare.22664044. The remaining data supporting the findings of this study are available within the Article, Supplementary Information or Source Data file. Source data are provided with this paper.

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

## Acknowledgements

This work was supported by National Key R&D Program of China (No. 2021YFA0909900), National Natural Science Foundation of China (Nos. 82073777 and 81803442) and General Project of Liaoning Provincial Department of Education (Nos. LJKZ0927 and LJKQZ2021034) and Natural Science Foundation of Liaoning Province (No. 2022-BS-157).

## Author contributions

Conceptualization: J.S., F.L., M.S., Z.H. Methodology: J.S., W.B., M.S., F.L., H.H. Performed all the experiments: W.B., M.S., W.H., S.P., B.L., Z.C. Analyzed all data: W.B., M.S., P.L. Writing-review and editing: all authors.

## Competing interests

The authors declare no competing interests.

## Additional information

¹Wuya College of Innovation, Shenyang Pharmaceutical University, Shenyang 110016 Liaoning, China. ²School of Pharmacy, Shenyang Pharmaceutical University, Shenyang 110016 Liaoning, China. ³Department of Surgical Oncology and General Surgery, The First Hospital of China Medical University, Key Laboratory of Precision Diagnosis and Treatment of Gastrointestinal Tumors, China Medical University, Ministry of Education, Shenyang 110001 Liaoning, China. ⁴Sino-British Research Centre for Molecular Oncology, National Centre for International Research in Cell and Gene Therapy, School of Basic Medical Sciences, Academy of Medical Sciences, Zhengzhou University, Zhengzhou 450052, China. ⁵These authors contributed equally: Weiyue Ban, Mengchi Sun, Hanwei Huang, Wanxu Huang. ✉e-mail: fnliu@cmu.edu.cn; sunjin@syphu.edu.cn

