## [Peer Review File · Nature Communications]

Engineered bacterial outer membrane vesicles encapsulating oncolytic adenoviruses enhance the efficacy of cancer virotherapy by augmenting tumor cell autophagyREVIEWER COMMENTS

Reviewer #1 (Remarks to the Author): with expertise in oncolytic viruses, cancer immunology

This manuscript reports on the use of Outer Membrane Vesicles derived from E-Coli and containing P2O as a delivery vehicle for an Adenovirus. The concept of using these OMV to shield the virus from immune neutralization is very interesting.

However, as written at the moment, the manuscript lacks sufficient experimental detail, is poorly written in terms of what experiments were done and how and the data do not address the mechanisms by which OMV protect Ad from, for example, neutralizing antibodies.

I have made some representative comments about how the first two Figures (Figures 2&3)- because there is no Figure 1- could be improved in terms of explanation, clarification, experimental detail and statistical analysis. Similar points can be raised for the remaining 4 Figures and for the additional 27 Supplemental Figures.

There is no Figure 1.

Figure 2A: Unclear as to what we are seeing and what we are supposed to be seeing in this Figure. How many Ad are encapsulated in the OMVs@P2O-Ads?

Figure 2C: SDS Page does not show specificity of the 70KDa P2O and the loading of the lanes is different.

Figure 2D: Experimental details need to be provided.

Figure 2E: Needs to be bigger and explanation of what the arrows are showing.

Figure 2i: needs experimental detail.

Figure 2J should be quantified with statistical analysis.

Figure 3A: Need to show multiple mice. Need experimental detail.

Figure 3C needs error bars for statistical relevance and the legend needs experimental detail.

Figure 3D: n=6 in the Legend is not reflected in the Figure where n=4.

And so on.

Reviewer #2 (Remarks to the Author): with expertise in oncolytic viruses, autophagy

Ban and colleagues provide a report detailing the construction of biomineral engineered OMVs-encapsulating oncolytic adenovirus that exhibit enhanced antitumor efficacy. It was mainly dependent on overactivated autophagy. Some areas where improvements can be made include:

Major concerns:

1. Fig 2f/4f: Grey-scale analysis should be performed, so that the LC3-II/LC3-I ratio can be calculated and statistically analyzed.
2. Fig 3f-k: Since immune response is a dynamic process, from innate immunity to T cell mediated

immunity and B cell mediated immunity, so it is critical to specify at what time point did they collect the tumor samples and explain why they choose this time point.

3. Fig 3j: Grouping information and FDR should also be presented in the GSEA figure. Unexpectedly, the pathway "Activation of immune response" is not included in Fig S11. The authors should explain the representativeness of choosing this pathway.

4. Fig 3k: The individual variations among the three tested samples in OMVs@P2O-Ads group are much too large. This kind of variation severely compromise the accuracy of the data.

5. Fig S11: Most of the GSEA enriched pathways are related to B cells, suggesting that the immune response induced by OMVs@P2O-Ads seems to be mediated by B cells, but not T cells. As far as we know, antitumor immunity is mostly mediated by T cells. Thus, it would be better for the authors to explain why they did not study B cell mediated immunity. Perhaps 18 days post Ads inoculation is too late to monitor the T cell immunity.

6. Fig S13/S14 are extremely important and should be presented in Fig 4. What kind of fluorescent dye did the authors used in Fig S13? This information should be provided in figure legends.

7. Fig 5c: Large amount of tiny green spots, which are unlikely to be normal CD8 staining, are shown in G5 and G6. The authors would be better to provide explanation.

8. Fig 5: The authors try to demonstrate that the antitumor efficacy of CaP-OMVs@P2O-Ads depend on the activation of CD8 T cells. In this case, more solid evidences, including the activation status of CD8 T cells (CD44 and CD69 expression), the tumor killing activity of CD8 T cells (co-culture assay), and dependency of CD8 T cells (depleting CD8 with antibodies), should be provided.

9. Fig 5d-f: The flow cytometry was not well performed. Large amount of death cells leads to serious unspecific staining, which adversely affect the interpretation and quantification of the data. The authors should use live/dead dyes to exclude death cells and debris.

10. Fig 5f: Since most, if not all, antigen presenting cells (APCs) express CD80 and CD86, these two markers are not specific enough to identify DCs. The authors should use CD11c and MHCII, instead.

11. Discussion section is missing in the current manuscript.

Minor concerns:

1. The language of the paper could be improved with some editing.

2. It would be better to have an introduction of the advantages and disadvantages of bacterial outer membrane vesicles. Are OMVs better than other nanomaterials? Are there any potential safety concerns?

3. Figure 1 is missing.

4. Fig 2g is missing.

5. Fig 2j: What do G1~G6 represent? The authors should mention this information in the figure legends.

6. Fig 2k should be mentioned at least once in the manuscript.

7. Fig S7: Only 6 columns are presented, but x-axis has 8 groups.

8. Fig S8/5b/6c: What does "Rr=6/6" or "Rr=5/5" mean?

9. Fig S10: What do G1~G6 represent here. Are they the same with Fig S9? The authors should mention this information in the figure legends.

10. The authors should explain why they use TC-1-hCD46. Indeed, hCD46 is the receptor for adenovirus.

11. Misspell: "wight" in section 2.2

12. Fig 3a: The authors ought to give a brief introduction of DiR dye.

13. Figure 4: It would be better for the authors to explain why CaP-OMVs exhibit better tumor selectivity than OMVs.

14. Fig 4b: What do C, O, P, S, Ca, and Ca+P stand for?

15. Fig S22-23: Gating strategies should also be presented.

16. Fig S9/S16/S17/S24/3c/: What do the dotted lines represent?

17. Section 2.5: Some OVVs in clinical trials, including vaccinia virus and reovirus, are systemically delivered. The authors should mention this and compare the CaP-OMVs technology with these intravenous OVVs.

18. Section 3: What dose "the oncolytic Ads extracted from E. coli" mean? Ads is grown in HEK293 cells?

Reviewer #3 (Remarks to the Author): with expertise in outer membrane vesicles, immunotherapy

In the current study, the authors design and develop a modified oncolytic adenovirus to address the intrinsic drawbacks of the virus. They used biomineral bacterial outer membrane vesicles encapsulated adenovirus to stimulate autophagy and antitumor immunity. The integrated immunotherapy is timely and critical for improving the clinical applications of the oncologic adenovirus and will attract significant attentions from broad readership. There are some important issues the authors should consider to clarify or improve in the revised version.

1. The logic to integrate various components is rather weak and it is recommended for the authors to clarify in the manuscript. Are these components are replaceable or necessary? It is a complicated system and it is hardly be treated as composite microbe. It is recommended to change the word with nanocomposite or nanosystem.
2. How the adenovirus loaded into OMV? What is the efficacy and any improvement have been tried?
3. Autophagy-overactivated is not proper expression, since overactivated action infers to uncontrolled process and may lead to severe side effects.
4. Quantitative measurement of pyranose oxidase in critical in vivo. What is the contribution for this enzyme for immune activation?
5. The scholarly presentation needs to further improve, such as no OV definition provided in the manuscript.
6. For the immune activation experiments, various critical steps are missing to generate a concrete conclusion of cascade antitumor activation.

Reviewer #4 (Remarks to the Author): with expertise in oncolytic viruses, autophagy, nanotherapy.

This is a meaningful work for the present autophagy-cascade-boosted immunotherapeutic method. The authors stated that OMVs@P2O

promoted Ads replication and resulted in Ads-overactivated autophagy, further remolded immunosuppressive TME. However, several problems that must be clarified need to be solved.

1. As we all known, oncolytic adenovirus enters tumor cells through CAR receptor to play an anti-tumor role. What mechanism does OMVs@P2O or OMVs@P2O-Ads enter tumor cells through? Does it have practical significance in tumor cells with high or low CAR expression?
2. The reason of the low intratumoral content of intravenous-delivered Ads is that the higher level of anti-adenovirus antibody in human body eliminates the exogenous injected Ads. Can OMVs@P2O or OMVs@P2O-Ads effectively avoid the elimination of neutralizing antibodies? Whether the expression level of anti-adenovirus antibody has been improved in the mouse model in advance? This is a very necessary experiment.
3. Infection with oncolytic viruses leads to activation of type I IFN signaling pathways, which are crucial in oncolytic virus-mediated antitumor immunity. The authors stated that OMVs@P2O promoted Ads replication. Is this pathway activated to a greater extent by OMVs@P2O?
4. In vivo experiment on OMVs@P2O-Ads or CaP-OMVs@P2O-Ads regulating tumor immune microenvironment is not enough. The innate and adaptive immune cells, as well as the activation and exhausted markers of T cells, need to be detected.

To sum up, my review opinion is that unless the authors can completely and effectively supplement the above experiments, it is unacceptable.

**Responses to the reviewers' comments**

**Reviewer #1:** This manuscript reports on the use of Outer Membrane Vesicles derived from *E-Coli*
and containing P₂O as a delivery vehicle for an Adenovirus. The concept of using these OMV to
shield the virus from immune neutralization is very interesting. However, as written at the moment,
the manuscript lacks sufficient experimental detail, is poorly written in terms of what experiments
were done and how and the data do not address the mechanisms by which OMV protect Ad from,
for example, neutralizing antibodies. I have made some representative comments about how the
first two Figures (Figures 2&3)- because there is no Figure 1- could be improved in terms of
explanation, clarification, experimental detail and statistical analysis. Similar points can be raised
for the remaining 4 Figures and for the additional 27 Supplemental Figures.

**Question 1:** There is no Figure 1.

**Response:** We are sorry that Figure 1 had been not shown in the manuscript. We have attached
Figure 1 here and added it to the revised manuscript (page 6).

**Figure 1. Schematic diagram.** The biomaterialized microbial nanocomposite engineered from OVVs for
 autophagy-cascade-augmented immunotherapy.

**Question 2:** Figure 2A: Unclear as to what we are seeing and what we are supposed to be seeing in
 this Figure. How many Ad are encapsulated in the OMVs@ P_2O -Ads?

**Response:** Thanks for the reviewer's kind questions. Figure 2A showed the particle size and size
 distribution of Ads, OMVs@ P_2O and OMVs@ P_2O -Ads (measured by Malvern laser granulometer),
 as well as the morphology of these under the transmission electron microscope (TEM). In TEM
 images, Ads possessed a hexagonal core, and OMVs@ P_2O presented a spherical shell. The
 "core-shell" structure of OMVs@ P_2O -Ads had been shown in Figure 2A, indicating that the
 successful construction of this microbial nanocomposite. In this study, Ads themselves have a

steady particle size of 90-100 nm. The *E. coli*-secreted OMVs possessed an even size about 130 nm
 through the extraction method. Thus, we ensure that each microbial nanocomposite contained only
 one Ad particle, which are consistent with image of TEM.

**Figure 2. Preparation and *in vitro* evaluation of the microbial nanocomposite.** (a) TEM and size distribution
images of Ads, OMVs@P₂O, and OMVs@P₂O-Ads. Scale bar=100 nm. (b) CLSM images of the microbial
nanocomposite. Ads were stained with DAPI dye (red) and OMVs carried a GFP marker (green). Scale bar=1 μm.
(c) The expression of P₂O was investigated by the SDS-PAGE method. (d) The ROS level assessment in TC-1
cells by flow cytometry. (e) TEM images of autophagosomes. Scale bar=200 nm. (f) The expression of
autophagy-related protein LC3-I and LC3-II by western bolt analyses. (g) CLSM images of autophagosomes.
Cells were stained with EB dye (red) and autophagosomes were stained with MDC dye (blue). Scale bar=50 μm.
(h) The Ads replication in TC-1 cells was quantified using real-time PCR at 0, 24, 36, and 48 h sequentially. 3MA
is an autophagy inhibitor: 3-Methyladenine. (i) Cytotoxicity of different formulations in TC-1 cells by CLSM.
Living cells were stained with Calcein (green) and dead cells were stained with PI (red). Scale bar=20 μm. (j)
Schematic diagram of bridging ROS with oncolytic Ads replication. **p*<0.05, ***p*<0.01, ****p*<0.001,
*****p*<0.0001 versus control. G1: PBS, G2: Ads, G3: OMVs, G4: OMVs@P₂O, G5: OMVs-Ads, G6:
OMVs@P₂O-Ads.

**Question 3:** Figure 2C: SDS Page does not show specificity of the 70 KDa P₂O and the loading of
the lanes is different.

**Response:** We appreciate the reviewer's comments. Pyranose oxidase (P₂O) is an enzyme with a
molecular mass of 70KDa. As shown in Figure 2C, compared with OMVs group, there are clearer
70KDa lanes observed in OMVs@P₂O, OMVs@P₂O-Ads and *E. coli*@P₂O groups. However, due
to the limitations of SDS Page method, the expression of the P₂O can't be specifically confirmed.
So, in our study, a chemical chromogenic reaction approach was conducted to further verify the
function of the P₂O, thereby indirectly proving the existence of P₂O (Figure S1- S4).

In our study, to ensure the rigor of SDS experiments, the total protein content of the samples from
all groups was measured by BCA protein quantification kit and the loading quantity of all groups
were kept in a consistent value. Here, we had re-modified experiments and obtained experimental
results in the revised manuscript as follow (Figure 2C):

**Figure S1.** Qualitative analysis of P₂O expression in OMVs, *E. coli*@P₂O and OMVs@P₂O.

Figure S2. Qualitative analysis of P₂O expression in OMVs@P₂O-Ads.

Figure S3. Qualitative analysis of different concentration P₂O expression in OMVs@P₂O.

**Figure S4.** The function curve illustrated the relationship between the absorbance and P₂O with different
concentrations. The red point revealed the relative P₂O concentration within the microbial nanocomposite.

**Figure 2. Preparation and *in vitro* evaluation of the microbial nanocomposite.** (a) TEM and size distribution
 images of Ads, OMVs@P₂O, and OMVs@P₂O-Ads. Scale bar=100 nm. (b) CLSM images of the microbial
 nanocomposite. Ads were stained with DAPI dye (red) and OMVs carried a GFP marker (green). Scale bar=1 μm.
 (c) The expression of P₂O was investigated by the SDS-PAGE method. (d) The ROS level assessment in TC-1

cells by flow cytometry. (e) TEM images of autophagosomes. Scale bar=200 nm. (f) The expression of
autophagy-related protein LC3-I and LC3-II by western bolt analyses. (g) CLSM images of autophagosomes.
Cells were stained with EB dye (red) and autophagosomes were stained with MDC dye (blue). Scale bar=50 μm .
(h) The Ads replication in TC-1 cells was quantified using real-time PCR at 0, 24, 36, and 48 h sequentially. 3MA
is an autophagy inhibitor: 3-Methyladenine. (i) Cytotoxicity of different formulations in TC-1 cells by CLSM.
Living cells were stained with Calcein (green) and dead cells were stained with PI (red). Scale bar=20 μm . (j)
Schematic diagram of bridging ROS with oncolytic Ads replication. * $p<0.05$, ** $p<0.01$, *** $p<0.001$,
**** $p<0.0001$ versus control. G1: PBS, G2: Ads, G3: OMVs, G4: OMVs@P₂O, G5: OMVs-Ads, G6:
OMVs@P₂O-Ads.

**Question 4:** Figure 2D: Experimental details need to be provided.

**Response:** We appreciate the reviewer's comments. In the revised manuscript, this part was
modified in section 4.6 as (page 23): "TC-1 mouse lung cancer cells were cultured in DMEM
medium supplemented with 10% FBS, penicillin (100 U mL⁻¹), and streptomycin (100 μg mL⁻¹).
The cells were maintained in a humidified atmosphere of 5% CO₂ at 37 °C. TC-1 cells were
cultured and divided into three groups (PBS, OMVs, and OMVs@P₂O). After the cells occupied
80% of the bottom, the medium was discarded, and the cells were rinsed twice using PBS.
DCFH-DA fluorescent dye (10 μM , 1mL) (Meilun ROS Assay Kit MA0219) was added to the
blank medium working solution. The cells were then incubated at 37 °C for 1 h in the dark. Next,
the medium was discarded, and the cells were rinsed with PBS again. Then, PBS, OMVs, and
OMVs@P₂O blank medium dispersion were added sequentially and further incubated at 37°C for 3
81 h in the dark. After rinsing with PBS, the cells were collected, and their ROS concentration was
82 determined by flow cytometry. The extracellular DCFH-DA has no fluorescence even after
83 possessing the capability of crossing the cell membrane freely. After entering the cell, it can be
hydrolyzed by intracellular esterase to translate into DCFH, which cannot pass through the cell
membrane. In the presence of ROS, DCFH is oxidized to produce the fluorescent substance DCF
(the excitation wavelength: 502nm; the emission wavelength: 530nm)."

**Question 5:** Figure 2E: Needs to be bigger and explanation of what the arrows are showing.

**Response:** We appreciate the reviewer's comments. In order to make this article more intuitive for
reviewers and readers, we tried to put the equal-scaling amplifying pictures in the supplement
(Figure S7).

**Figure S7.** The equal-scaling amplifying TEM images of autophagosomes, scale bar=500 nm.

**Question 6:** Figure 2i: needs experimental detail.

**Response:** We appreciate the reviewer's comments. In the revised manuscript, this part (replaced
 by Figure 2h) was modified in section 4.9 as (page 25): "TC-1 cells were cultured in cell culture
 plates. After the cells occupied 80% of the bottom, the medium was sucked out with a 2 mL syringe
 and washed with PBS twice. Ads, OMVs-Ads, OMVs@P₂O-Ads + 3MA, and OMVs@P₂O-Ads
 were added to each group and incubated for 3h at 37°C. 3MA is an autophagy inhibitor:
 3-Methyladenine. The drug solution was dumped out, the cells were rinsed twice with PBS, and the
 same volume of blank medium was added. Plates were put in 37 °C and moved after 0, 24 h, 36 h,
 and 48 h, and were placed at -80°C for three times to ensure complete cell breakdown. The
 freeze-thaw solution was collected and centrifuged at 2800 rpm/min at 4°C for 0.5 h. Precipitation
 was discarded, and 1% triton was added to the supernatant. Then, the liquid was pre-denatured at
 98°C, and the precipitate of deformed protein was removed by centrifugation at 12000 rpm/min for
 10 min. Finally, the RT-qPCR technique collected and processed the supernatant for quantitative
 Ads detection. All the reagents of the RT-qPCR technique were purchased from Vazyme."

**Question 7:** Figure 2J should be quantified with statistical analysis.

**Response:** We appreciate the reviewer's comments. The statistical analysis result had been shown
in Figure S10 in the revised manuscript:

**Figure S10.** The statistical analysis result of live/dead cellular staining (n=3). (G1: PBS, G2: Ads, G3: OMVs, G4:
OMVs@P₂O, G5: OMVs-Ads, G6: OMVs@P₂O-Ads).

**Question 8:** Figure 3A: Need to show multiple mice. Need experimental detail.

**Response:** We agree with the reviewer's comments. The more mice are shown below. In the revised
manuscript, this part was modified in section 4.11 as (page 25):

“Female C57 mice were obtained from the Laboratory Animal Center of Shenyang
Pharmaceutical University. The animal experiments were performed by following the Guidelines
for the Care and Use of Laboratory Animals approved by the Institutional Animal Ethical Care
Committee (IAEC) of Shenyang Pharmaceutical University.

OMVs@P₂O-Ads was prepared as described in section 4.4, and an excess of DIR staining
solution was subsequently added to label OMVs. The free DIR dye was removed by centrifugation
at 3,000 × g for 3 min using an ultrafiltration tube with a 100 kDA pore size. DIR-labeled
OMVs@P₂O-Ads were injected intratumorally using Ads content of 7 × 10⁵ PFU as a standard. DIR
fluorescent imaging of the microbial nanocomposite *in vivo* in TC-1-hCD46 xenograft
tumor-bearing mice by IVIS.”

**Figure S11.** *In vivo* DIR fluorescent imaging of the nanocomposite in TC-1-hCD46 xenograft tumor-bearing mice
 by IVIS (n=3).

**Question 9:** Figure 3C needs error bars for statistical relevance and the legend needs experimental
 detail.

**Response:** We appreciate the reviewer's comments. Error bars are represented by dotted lines in
 Figure 3C. To make the figure information more intuitive for reviewers and readers, we replace the
 dotted lines with the traditional error bars. Furthermore, the experimental detail was attached as
 follow (page 12):

**Figure 3. *In vivo* oncolytic efficacy of the microbial nanocomposite.** (a) *In vivo* DIR fluorescent imaging of the
 microbial nanocomposite in TC-1-hCD46 xenograft tumor-bearing mice by IVIS. (b) Schematic illustration of the
 antitumor activity and immunology assessment experiments of the microbial nanocomposite using TC-1-hCD46
 xenograft tumor-bearing C57 female mice model. TC-1 cells (10^6) were subcutaneously injected into the waist of
 female C57 mice, and the tumor-bearing mice were divided into six groups ($n=6$). When the tumor reached
 100-150 mm^3 , the mice were injected intratumorally with PBS, Ads (7×10^5 PFU), OMVs, OMVs@P₂O,
 OMVs-Ads (7×10^5 PFU), and OMVs@P₂O-Ads (7×10^5 PFU). The drug was given every three days for four
 consecutive times, the tumor volume was measured with a vernier caliper, and mice were weighed daily. (c)
 Tumor volume growth profiles of C57 mice bearing TC-1 xenografts. (d) Images of representative tumors of
 different treated groups on the 18th day ($n=6$). (e) Statistical graph of tumor weight of different treated groups
 on the 18th day ($n=6$). (f-h) Images of concentration of main cytokines in serum. (i) The differential gene expression
 between the samples treated with OMVs@P₂O-Ads and PBS, using the absolute value of logFC greater than 1 as

the threshold. (j) GSEA enriched pathways of the up-regulated genes in the samples treated with
 OMVs@P₂O-Ads, showing immune-related terms. (k) Gene set enrichment analysis (GSEA) of the term
 “Activation of immune response”, and the genes included in this pathway are highlighted in (i) with light yellow
 brown. **p*<0.05, ***p*<0.01, ****p*<0.001, *****p*<0.0001 versus control. (G1: PBS, G2: Ads, G3: OMVs, G4:
 OMVs@P₂O, G5: OMVs-Ads, G6: OMVs@P₂O-Ads).

**Question 10:** Figure 3D: n=6 in the Legend is not reflected in the Figure where n=4.

**Response:** We are sorry that we made an error in the process of typesetting, which resulted in
 Figure 3D not being fully presented. We have attached the original documents here and modified in
 the revised manuscript.

**Figure 3. *In vivo* oncolytic efficacy of the microbial nanocomposite.** (a) *In vivo* DIR fluorescent imaging of the
microbial nanocomposite in TC-1-hCD46 xenograft tumor-bearing mice by IVIS. (b) Schematic illustration of the
antitumor activity and immunology assessment experiments of the microbial nanocomposite using TC-1-hCD46
xenograft tumor-bearing C57 female mice model. TC-1 cells (10^6) were subcutaneously injected into the waist of
female C57 mice, and the tumor-bearing mice were divided into six groups (n=6). When the tumor reached
100-150 mm³, the mice were injected intratumorally with PBS, Ads (7×10^5 PFU), OMVs, OMVs@P₂O,
OMVs-Ads (7×10^5 PFU), and OMVs@P₂O-Ads (7×10^5 PFU). The drug was given every three days for four
consecutive times, the tumor volume was measured with a vernier caliper, and mice were weighed daily. (c)
Tumor volume growth profiles of C57 mice bearing TC-1 xenografts. (d) Images of representative tumors of
different treated groups on the 18th day (n=6). (e) Statistical graph of tumor weight of different treated groups
on the 18th day (n=6). (f-h) Images of concentration of main cytokines in serum. (i) The differential gene expression
between the samples treated with OMVs@P₂O-Ads and PBS, using the absolute value of logFC greater than 1 as
the threshold. (j) GSEA enriched pathways of the up-regulated genes in the samples treated with
OMVs@P₂O-Ads, showing immune-related terms. (k) Gene set enrichment analysis (GSEA) of the term
“Activation of immune response”, and the genes included in this pathway are highlighted in (i) with light yellow
brown. * $p < 0.05$, ** $p < 0.01$, *** $p < 0.001$, **** $p < 0.0001$ versus control. (G1: PBS, G2: Ads, G3: OMVs, G4:
OMVs@P₂O, G5: OMVs-Ads, G6: OMVs@P₂O-Ads).

**Reviewer #2:** Ban and colleagues provide a report detailing the construction of biomineral
engineered OMVs-encapsulating oncolytic adenovirus that exhibit enhanced antitumor efficacy. It
was mainly dependent on overactivated autophagy. Some areas where improvements can be made
include:

Major concerns:

**Question 1:** Fig 2f/4f: Grey-scale analysis should be performed, so that the LC3-II/LC3-I ratio can
be calculated and statistically analyzed.

**Response:** We appreciate the reviewer's comments. The grey-scale analysis in Figure2f/4f had been
conducted, and LC3-II/LC3-I ratio had been calculated and statistically analyzed in the revised
manuscript.

**Figure 2. Preparation and *in vitro* evaluation of the microbial nanocomposite.** (a) TEM and size distribution
 images of Ads, OMVs@P₂O, and OMVs@P₂O-Ads. Scale bar=100 nm. (b) CLSM images of the microbial
 nanocomposite. Ads were stained with DAPI dye (red) and OMVs carried a GFP marker (green). Scale bar=1 μm.
 (c) The expression of P₂O was investigated by the SDS-PAGE method. (d) The ROS level assessment in TC-1

cells by flow cytometry. (e) TEM images of autophagosomes. Scale bar=200 nm. (f) The expression of
autophagy-related protein LC3-I and LC3-II by western bolt analyses. (g) CLSM images of autophagosomes.
Cells were stained with EB dye (red) and autophagosomes were stained with MDC dye (blue). Scale bar=50 μ m.
(h) The Ads replication in TC-1 cells was quantified using real-time PCR at 0, 24, 36, and 48 h sequentially. 3MA
is an autophagy inhibitor: 3-Methyladenine. (i) Cytotoxicity of different formulations in TC-1 cells by CLSM.
Living cells were stained with Calcein (green) and dead cells were stained with PI (red). Scale bar=20 μ m. (j)
Schematic diagram of bridging ROS with oncolytic Ads replication. * p <0.05, ** p <0.01, *** p <0.001,
**** p <0.0001 versus control. G1: PBS, G2: Ads, G3: OMVs, G4: OMVs@P₂O, G5: OMVs-Ads, G6:
OMVs@P₂O-Ads.

Figure S8. The LC3-II/LC3-I ratio *in vitro* (n=3).

**Figure 4. Preparation and *in vivo* evaluation of the biom mineralized microbial nanocomposite.** (a) TEM and
 size distribution images of Ads, OMVs@P₂O-Ads, and CaP-OMVs@P₂O-Ads. Scale bar=100 nm. (b) Energy

spectrum analysis image of the biom mineralized composite microbe. Scale bar=50 nm. (c) *In vivo* fluorescence
 imaging of the multiple organs and tumors collected from the mice at 24 h post *i.v.* injection. From left to right:
 tumor, heart, liver, spleen, lung, and kidney. (d) Quantitation of the biodistribution of relative Ads contents in
 multiple organs and tumors after 24 h of different treatments by RT-qPCR (n=4). (e) Immunofluorescence images
 of LC3 autophagic proteins in tumor tissues. Blue represents DAPI-stained tumor cells and the green represents
 FITC-stained LC3 autophagic protein. Scale bar=2 mm. (f) Quantitative analysis of fluorescence intensity. (g) The
 expression of autophagy-related protein LC3-I and LC3-II examined by western blot. (h) Quantitation of relative
 Ads content in the tumor after 72 h of different treatments by RT-qPCR technique (n=3). *p<0.05, **p<0.01,
 ***p<0.001, ****p<0.0001 versus control. *p<0.05, **p<0.01, ***p<0.001, ****p<0.0001 versus control.

**Figure18.** The LC3-II/LC3-I ratio *in vivo* (n=3).

**Question 2:** Fig 3f-k: Since immune response is a dynamic process, from innate immunity to T cell
 mediated immunity and B cell mediated immunity, so it is critical to specify at what time point did
 they collect the tumor samples and explain why they choose this time point.

**Response:** We appreciate the reviewer's comments. As for Figure 3f-h, the collection of samples
 for immunological studies was performed on the basis of pharmacodynamic studies. Concretely, as
 shown in Figure 3b, the mice were injected different drugs at 1, 4, 7 and 10 days and dissected at
 the 18th day. And as for Figure 3i-k, the collection of samples for transcriptomic analysis of the
 tumor xenografts 7 days after the first administration. Compared with the expression of cytokines
 and the visualization of tumor volume, relevant transcriptomic change of the tumor xenografts is
 earlier, this is why we accomplished the transcriptomic analysis of the tumor xenografts after two
 consecutive administration (on the seventh day). However, as described by the reviewer's

**Question2 - Question5,** we have been aware that the collection of samples for transcriptomic
 analysis of the tumor xenografts at 7 day is too early. In the revised manuscript, we redesigned the

experiment and collected samples for transcriptomic analysis on the twelfth day after the fourth
 administration. And the relevant results are shown in Figure 3i-k.

**Figure 3. In vivo oncolytic efficacy of the microbial nanocomposite.** (a) *In vivo* DIR fluorescent imaging of the
 microbial nanocomposite in TC-1-hCD46 xenograft tumor-bearing mice by IVIS. (b) Schematic illustration of the
 antitumor activity and immunology assessment experiments of the microbial nanocomposite using TC-1-hCD46
 xenograft tumor-bearing C57 female mice model. TC-1 cells (10⁶) were subcutaneously injected into the waist of
 female C57 mice, and the tumor-bearing mice were divided into six groups (n=6). When the tumor reached
 100-150 mm³, the mice were injected intratumorally with PBS, Ads (7×10⁵ PFU), OMVs, OMVs@P₂O,
 OMVs-Ads (7×10⁵ PFU), and OMVs@P₂O-Ads (7×10⁵ PFU). The drug was given every three days for four
 consecutive times, the tumor volume was measured with a vernier caliper, and mice were weighed daily. (c)
 Tumor volume growth profiles of C57 mice bearing TC-1 xenografts. (d) Images of representative tumors of

different treated groups on the 18th day (n=6). (e) Statistical graph of tumor weight of different treated groups on
the 18th day (n=6). (f-h) Images of concentration of main cytokines in serum. (i) The differential gene expression
between the samples treated with OMVs@P₂O-Ads and PBS, using the absolute value of logFC greater than 1 as
the threshold. (j) GSEA enriched pathways of the up-regulated genes in the samples treated with
OMVs@P₂O-Ads, showing immune-related terms. (k) Gene set enrichment analysis (GSEA) of the term
“Activation of immune response”, and the genes included in this pathway are highlighted in (i) with light yellow
brown. **p*<0.05, ***p*<0.01, ****p*<0.001, *****p*<0.0001 versus control. (G1: PBS, G2: Ads, G3: OMVs, G4:
OMVs@P₂O, G5: OMVs-Ads, G6: OMVs@P₂O-Ads).

**Question 3:** Fig 3j: Grouping information and FDR should also be presented in the GSEA figure.
Unexpectedly, the pathway “Activation of immune response” is not included in Fig S11. The
authors should explain the representativeness of choosing this pathway.

**Response:** Thanks for this kind suggestion, and grouping information and FDR had been added in
the GSEA figure (Fig. 3k) in the revised manuscript. The original Fig S11 was ranked by the P
value and also limited terms were shown, thus the pathway “Activation of immune response” was
not included. In the revised manuscript, we improved the time-point to acquire the samples and
re-performed the transcriptomic analysis and showed that the pathway “Activation of immune
response” is significantly up-regulated 12 days after OMVs@P₂O-Ads injection (Fig 3j).

**Figure 3. *In vivo* oncolytic efficacy of the microbial nanocomposite.** (a) *In vivo* DIR fluorescent imaging of the
 microbial nanocomposite in TC-1-hCD46 xenograft tumor-bearing mice by IVIS. (b) Schematic illustration of the
 antitumor activity and immunology assessment experiments of the microbial nanocomposite using TC-1-hCD46
 xenograft tumor-bearing C57 female mice model. TC-1 cells (10^6) were subcutaneously injected into the waist of
 female C57 mice, and the tumor-bearing mice were divided into six groups ($n=6$). When the tumor reached
 $100\text{-}150\text{ mm}^3$, the mice were injected intratumorally with PBS, Ads (7×10^5 PFU), OMVs, OMVs@P₂O,
 OMVs-Ads (7×10^5 PFU), and OMVs@P₂O-Ads (7×10^5 PFU). The drug was given every three days for four
 consecutive times, the tumor volume was measured with a vernier caliper, and mice were weighed daily. (c)
 Tumor volume growth profiles of C57 mice bearing TC-1 xenografts. (d) Images of representative tumors of
 different treated groups on the 18th day ($n=6$). (e) Statistical graph of tumor weight of different treated groups
 on the 18th day ($n=6$). (f-h) Images of concentration of main cytokines in serum. (i) **The differential gene expression**
 **between the samples treated with OMVs@P₂O-Ads and PBS, using the absolute value of logFC greater than 1 as**

the threshold. (j) GSEA enriched pathways of the up-regulated genes in the samples treated with
OMVs@P₂O-Ads, showing immune-related terms. (k) Gene set enrichment analysis (GSEA) of the term
“Activation of immune response”, and the genes included in this pathway are highlighted in (i) with light yellow
brown. * $p < 0.05$, ** $p < 0.01$, *** $p < 0.001$, **** $p < 0.0001$ versus control. (G1: PBS, G2: Ads, G3: OMVs, G4:
OMVs@P₂O, G5: OMVs-Ads, G6: OMVs@P₂O-Ads).

**Question 4:** Fig 3k: The individual variations among the three tested samples in OMVs@P₂O-Ads
group are much too large. This kind of variation severely compromise the accuracy of the data.

**Response:** Thanks for this kind suggestion. In our original experiment, the samples were acquired
earlier (on the seventh day), so that the immune response in some mice had not been invoked.
According to the reviewer’s suggestion and the result of our phenotypic experiment, samples were
acquired uniformly 11 days after OMVs@P₂O-Ads injection. And the experimental results obtained
according to the modified experimental plan are shown in the Figure 3i-k in the revised manuscript.

**Figure 3. *In vivo* oncolytic efficacy of the microbial nanocomposite.** (a) *In vivo* DIR fluorescent imaging of the
 microbial nanocomposite in TC-1-hCD46 xenograft tumor-bearing mice by IVIS. (b) Schematic illustration of the
 antitumor activity and immunology assessment experiments of the microbial nanocomposite using TC-1-hCD46
 xenograft tumor-bearing C57 female mice model. TC-1 cells (10^6) were subcutaneously injected into the waist of
 female C57 mice, and the tumor-bearing mice were divided into six groups ($n=6$). When the tumor reached
 100-150 mm^3 , the mice were injected intratumorally with PBS, Ads (7×10^5 PFU), OMVs, OMVs@P₂O,
 OMVs-Ads (7×10^5 PFU), and OMVs@P₂O-Ads (7×10^5 PFU). The drug was given every three days for four
 consecutive times, the tumor volume was measured with a vernier caliper, and mice were weighed daily. (c)
 Tumor volume growth profiles of C57 mice bearing TC-1 xenografts. (d) Images of representative tumors of
 different treated groups on the 18th day ($n=6$). (e) Statistical graph of tumor weight of different treated groups
 on the 18th day ($n=6$). (f-h) Images of concentration of main cytokines in serum. (i) **The differential gene expression**
 **between the samples treated with OMVs@P₂O-Ads and PBS, using the absolute value of logFC greater than 1 as**

the threshold. (j) GSEA enriched pathways of the up-regulated genes in the samples treated with
OMVs@P₂O-Ads, showing immune-related terms. (k) Gene set enrichment analysis (GSEA) of the term
“Activation of immune response”, and the genes included in this pathway are highlighted in (i) with light yellow
brown. * $p < 0.05$, ** $p < 0.01$, *** $p < 0.001$, **** $p < 0.0001$ versus control. (G1: PBS, G2: Ads, G3: OMVs, G4:
OMVs@P₂O, G5: OMVs-Ads, G6: OMVs@P₂O-Ads).

**Question 5:** Fig S11: Most of the GSEA enriched pathways are related to B cells, suggesting that
the immune response induced by OMVs@P₂O-Ads seems to be mediated by B cells, but not T cells.
As far as we know, antitumor immunity is mostly mediated by T cells. Thus, it would be better for
the authors to explain why they did not study B cell mediated immunity. Perhaps 18 days post Ads
inoculation is too late to monitor the T cell immunity.

**Response:** Thanks for this kind suggestion. Looking at the latest research progress of
microbe-mediated tumor immunotherapy, although the role of B cells in anti-tumor immunity is
gradually being discovered, as mentioned by the reviewer, anti-tumor immunity is mainly mediated
by T cells^{1, 2, 3}. In the experimental results of mice tumor transcriptome analysis shown in the
revised Figure 3j, most of the GSEA enriched pathways are related to B cells, which is consistent
with the results of other experiments (such as the increase of serum IL-6 in Fig 3g, which is capable
of promoting the differentiation of B cell) and is foreseen. Based on the experimental data of this
project and related literature reports, we believe that the activation of the B cell-associated
transcriptome is mainly caused by antiviral immunity instead of anti-tumor immunity. Although the
amplified anti-tumor immunity after injection of OMVs@P₂O-Ads is what we expect, as was
reviewed in our previous work, the occurrence of antiviral immunity was earlier and stronger than
anti-tumor immunity and the number of virus particles free in the tumor microenvironment is much
more than the number of viruses infected into the tumor cells during the whole immunity process⁴.
In the initial stage, free virions are mainly engulfed and eliminated by macrophages. After the
activation of specific antiviral immunity, B cell-mediated humoral immunity is mainly responsible
for the elimination of free Ads in tumor fluids. Overall, compared with innate immune cells and
specific antiviral T cells, B cells play a more significant role during the process of virus clearance,
which is the main reason why B cell-associated transcriptomes was distinctly activated as is shown
in the revised Figure 3j (the Fig S11 was removed to Figure 3j). However, in this study, we
attempted to focus on the anti-tumor immune response triggered by OMVs@P₂O-Ads instead of

antiviral immunity. Therefore, there is no doubt that it's more significant for us to meticulously
investigate the role of T cells in anti-tumor immunity process in our manuscripts even though most
GSEA-enriched pathways are associated with B cells. In addition, we agree with the reviewer's
opinion that perhaps 18 days post Ads inoculation is too late. In the revised manuscript, we have
re-modified the experiment and the transcriptome analysis of tumor tissues was performed on day
11, and the new results had been presented in Figure 3i-k in the revised manuscript. As shown in
Figure 3i-k, although some GSEA-enriched pathways were associated with B cells,
T-cell-associated GSEA-enriched pathways were also detected, indicating that the earlier detection
time points (on day 11 after the first administration) was more proper to investigate the change
situation of transcriptome in tumor tissue of mice, and the microbial nanocomposite
OMVs@P₂O-Ads possessed the ability to invoke T cell-mediated antitumor immunity.

**Figure 3. *In vivo* oncolytic efficacy of the microbial nanocomposite.** (a) *In vivo* DIR fluorescent imaging of the
microbial nanocomposite in TC-1-hCD46 xenograft tumor-bearing mice by IVIS. (b) Schematic illustration of the
antitumor activity and immunology assessment experiments of the microbial nanocomposite using TC-1-hCD46
xenograft tumor-bearing C57 female mice model. TC-1 cells (10^6) were subcutaneously injected into the waist of
female C57 mice, and the tumor-bearing mice were divided into six groups ($n=6$). When the tumor reached
100-150 mm^3 , the mice were injected intratumorally with PBS, Ads (7×10^5 PFU), OMVs, OMVs@P₂O,
OMVs-Ads (7×10^5 PFU), and OMVs@P₂O-Ads (7×10^5 PFU). The drug was given every three days for four
consecutive times, the tumor volume was measured with a vernier caliper, and mice were weighed daily. (c)
Tumor volume growth profiles of C57 mice bearing TC-1 xenografts. (d) Images of representative tumors of
different treated groups on the 18th day ($n=6$). (e) Statistical graph of tumor weight of different treated groups
on the 18th day ($n=6$). (f-h) Images of concentration of main cytokines in serum. (i) **The differential gene expression**
**between the samples treated with OMVs@P₂O-Ads and PBS, using the absolute value of logFC greater than 1 as**

the threshold. (j) GSEA enriched pathways of the up-regulated genes in the samples treated with
OMVs@P₂O-Ads, showing immune-related terms. (k) Gene set enrichment analysis (GSEA) of the term
“Activation of immune response”, and the genes included in this pathway are highlighted in (i) with light yellow
brown. * $p < 0.05$, ** $p < 0.01$, *** $p < 0.001$, **** $p < 0.0001$ versus control. (G1: PBS, G2: Ads, G3: OMVs, G4:
OMVs@P₂O, G5: OMVs-Ads, G6: OMVs@P₂O-Ads).

**References**

- 1. Cabrita R, et al. Tertiary lymphoid structures improve immunotherapy and survival in
melanoma. *Nature* 577, 561-565 (2020).
- 2. Petitprez F, et al. B cells are associated with survival and immunotherapy response in sarcoma.
*Nature* 577, 556-560 (2020).
- 3. Helmink BA, et al. B cells and tertiary lymphoid structures promote immunotherapy response.
*Nature* 577, 549-555 (2020).
- 4. Ban W, et al. Emerging systemic delivery strategies of oncolytic viruses: A key step toward
cancer immunotherapy. *Nano Res* 15, 4137-4153 (2022).

**Question 6:** Fig S13/S14 are extremely important and should be presented in Fig 4. What kind of
fluorescent dye did the authors used in Fig S13? This information should be provided in figure
legends.

**Response:** We appreciate the reviewer’s comments. In the revised manuscript, we have presented
Fig S13/S14 in Figure 4a and 4b. In Fig S13 (Figure 4a in the revised manuscript), we used
engineered eubacterial outer membrane vesicles that can express green fluorescent protein
(OMVs@GFP).

**Figure 4. Preparation and *in vivo* evaluation of the biom mineralized microbial nanocomposite.** (a) TEM and
 size distribution images of Ads, OMVs@P₂O-Ads, and CaP-OMVs@P₂O-Ads. Scale bar=100 nm. (b) Energy

spectrum analysis image of the biomineralized composite microbe. Scale bar=50 nm. (c) *In vivo* fluorescence
imaging of the multiple organs and tumors collected from the mice at 24 h post *i.v.* injection. From left to right:
tumor, heart, liver, spleen, lung, and kidney. (d) Quantitation of the biodistribution of relative Ads contents in
multiple organs and tumors after 24 h of different treatments by RT-qPCR (n=4). (e) Immunofluorescence images
of LC3 autophagic proteins in tumor tissues. Blue represents DAPI-stained tumor cells and the green represents
FITC-stained LC3 autophagic protein. Scale bar=2 mm. (f) Quantitative analysis of fluorescence intensity. (g) The
expression of autophagy-related protein LC3-I and LC3-II examined by western blot. (h) Quantitation of relative
Ads content in the tumor after 72 h of different treatments by RT-qPCR technique (n=3). * $p < 0.05$, ** $p < 0.01$,
*** $p < 0.001$, **** $p < 0.0001$ versus control. * $p < 0.05$, ** $p < 0.01$, *** $p < 0.001$, **** $p < 0.0001$ versus control.

**Question 7:** Fig 5c: Large amount of tiny green spots, which are unlikely to be normal CD8
staining, are shown in G5 and G6. The authors would be better to provide explanation.

**Response:** We appreciate the reviewer's comments. In original Figure 5c, the tiny green spots
indeed were CD8 staining, but the image quality was not satisfactory. In revised manuscript, we
modified the CD8 immunofluorescence section experiments using confocal fluorescence
microscopy, and the representative images and the fluorescence quantitative statistics are shown as
follows:

**Figure 5. *In vivo* oncolytic efficacy and immuneactivation capacity of the biomaterialized microbial**
 **nanocomposite.** (a) Schematic illustration of the antitumor activity and immunity investigation of the
 biomaterialized microbial nanocomposite on TC-1-hCD46 xenograft tumor-bearing C57 female mice model. (b)
 Individual tumor growth kinetics in different groups (n=6). (c) The immunofluorescence images of CD8⁺ T cells
 in tumor tissues. Scale bars=50μm. (d) Representative flow cytometric evolution images (g) as well as relative
 quantification of CD8⁺ T cells (CD45⁺CD3⁺CD8⁺) in the tumor (n=3). (e) Representative flow cytometric
 evolution images (h) as well as relative quantification of Treg cells (CD45⁺CD3⁺CD4⁺Fopx3⁺) in the tumor (n=3).
 (f) Representative flow cytometric evolution images (i) and relative quantification of MHC-II⁺ DC cells
 (CD45⁺CD11c⁺MHC-II⁺) in the tumor (n=3). **p*<0.05, ***p*<0.01, ****p*<0.001, *****p*<0.0001 versus control. (G1:

PBS, G2: OMVs@P₂O-Ads, G3: CaP-OMVs-Ads, G4: Intra-Ads, G5: CaP-OMVs@P₂O-Ads, G6: Intra-Ads high
does).

**Question 8:** Fig 5: The authors try to demonstrate that the antitumor efficacy of
CaP-OMVs@P₂O-Ads depend on the activation of CD8 T cells. In this case, more solid evidences,
including the activation status of CD8 T cells (CD44 and CD69 expression), the tumor killing
activity of CD8 T cells (co-culture assay), and dependency of CD8 T cells (depleting CD8 with
antibodies), should be provided.

**Response:** We appreciate the reviewer's comments. To refine the content of our experiments, we
supplemented the relevant experiments one by one as suggested by reviewer. First, we measured the
proportion of CD45⁺CD3⁺CD8⁺CD44⁺ and CD45⁺CD3⁺CD8⁺CD69⁺ T cells in tumor tissue after
four consecutive treatment of Cap-OMVs@P₂O-Ads *via* flow cytometry. As is shown in the Figure,
CD44⁻ T cells and CD44⁺ T cells clusters could be obviously observed. However, the cell cluster of
CD69⁺ T cells cannot be found in the figure, indicating that there is few T cell expressing CD69⁺
after four consecutive treatments of Cap-OMVs@P₂O-Ads. By reviewing the related papers, the
rationality of our experimental results was confirmed. Concretely, CD69 is one of the earliest
markers upregulated after T cell activation, whose expression increased in a time-dependent manner
between 3 and 12 hours, remained elevated until 24 hours, and then decreased¹. In our study, the
CD45⁺CD3⁺CD8⁺CD69⁺ T cells were measured after four consecutive treatment of
Cap-OMVs@P₂O-Ads. Therefore, we held the opinion that the T cells go through the primary
CD69⁺ activation phase and enter into the next activation stage.

**Figure.** Representative flow cytometric evolution image of CD45⁺CD3⁺CD8⁺CD44⁺ T cells and
 CD45⁺CD3⁺CD8⁺CD69⁺ T cells in tumor tissue.

Next, we have accomplished the co-culture assay *in vitro* to verify the tumor killing activity of
 CD8⁺ T cells in different administration groups. The detailed experimental method (section 4.23)
 and the experimental result (Figure S32) are as follows:

Method: “First, TC-1 cells (10^6) were subcutaneously injected into the waist of female C57 mice,
 and tumor-bearing mice were divided into six groups (n=6). When the tumor reached 100–150 mm³,
 the mice were intravenously injected with PBS, OMVs@P₂O-Ads, CaP-OMVs-Ads, and
 CaP-OMVs@P₂O-Ads, while the Ads (7×10^5 and 10^7) were injected intratumorally. CD8⁺ T cells
 were extracted from each administration group based on the instructions of the BeaverBeads™
 mouse CD8⁺ T cell sorting kit (purchased from Beaver, 70903-100). Then, TC-1 cells were cultured
 in cell culture plates. After the cells occupied 80% of the bottom, the medium was sucked out with a
 2 mL syringe and washed twice with PBS. Then, CD8⁺ T cells extracted from each administration
 group were added into the holes based on the proportion of TC-1 cells: CD8⁺ T cells =1:100 and
 subsequently incubated at 37°C for 24 h. Next, the MTT assay helped investigate the tumor-killing

rate in each group.”

**Figure S32.** The experimental result of the co-culture assay. (It's worth noting here that PBS represents T cells
extracted from mice in the PBS group, and other groups as above.)

In addition, the dependency of CD8 T cells was investigated *via* injecting CD8⁺ T cells antibody.
The detailed experimental method (section 4.24) and the experimental result (Figure S33) are as
follows:

Method: “TC-1 cells (10⁶) were subcutaneously injected into the waist of female C57 mice, and
tumor-bearing mice were divided into three groups (n=5). When the tumor reached the size of
100–150 mm³, the mice were intravenously injected with PBS, CaP-OMVs@P₂O-Ads, and
CaP-OMVs@P₂O-Ads plus CD8⁺ T cells antibody (anti-CD8 antibodies, clone: 2.43, Bio X cell,
cat. no.: BP0061, injected *i.v.* every two days starting one day before the CaP-OMVs@P₂O-Ads
injection) at 0, 3, 6, 9 day. On the 12th day of the efficacy experiment, the mice were sacrificed by
cervical spine removal, and the tumor tissue was isolated, weighed, and photographed. The effects
of the different preparations on tumor growth were analyzed and compared.”

**Figure S33.** Tumor volume during the treatments and images of representative tumors of different treated groups
on the 12th day (n=5).

**References**

- 1. Hamann J, Fiebig H, Strauss M. Expression cloning of the early activation antigen CD69, a
type II integral membrane protein with a C-type lectin domain. *J Immunol* 150, 4920-4927
(1993).

**Question 9:** Fig 5d-f: The flow cytometry was not well performed. Large amount of death cells
leads to serious unspecific staining, which adversely affect the interpretation and quantification of
the data. The authors should use live/dead dyes to exclude death cells and debris.

**Response:** We appreciate the reviewer's comments. Due to the interference of a large number of
dead cells, the Fig 5d-f data were not satisfactory. Therefore, we re-modified the experiment as
suggested by the reviewer including using live/dead dyes, and the results were shown below:

**Figure 5. *In vivo* oncolytic efficacy and immuneactivation capacity of the biomaterialized microbial**
 **nanocomposite.** (a) Schematic illustration of the antitumor activity and immunity investigation of the
 biomaterialized microbial nanocomposite on TC-1-hCD46 xenograft tumor-bearing C57 female mice model. (b)
 Individual tumor growth kinetics in different groups (n=6). (c) The immunofluorescence images of CD8⁺ T cells
 in tumor tissues. Scale bars=50 μm . (d) Representative flow cytometric evolution images (g) as well as relative
 quantification of CD8⁺ T cells (CD45⁺CD3⁺CD8⁺) in the tumor (n=3). (e) Representative flow cytometric
 evolution images (h) as well as relative quantification of Treg cells (CD45⁺CD3⁺CD4⁺Fopx3⁺) in the tumor (n=3).
 (f) Representative flow cytometric evolution images (i) and relative quantification of MHC-II⁺ DC cells
 (CD45⁺CD11c⁺MHC-II⁺) in the tumor (n=3). * $p < 0.05$, ** $p < 0.01$, *** $p < 0.001$, **** $p < 0.0001$ versus control. (G1:

PBS, G2: OMVs@P₂O-Ads, G3: CaP-OMVs-Ads, G4: Intra-Ads, G5: CaP-OMVs@P₂O-Ads, G6: Intra-Ads high
does).

**Question 10:** Fig 5f: Since most, if not all, antigen presenting cells (APCs) express CD80 and
CD86, these two markers are not specific enough to identify DCs. The authors should use CD11c
and MHCII, instead.

**Response:** We appreciate the reviewer's comments. In Figure 5f and 5i, we have replaced
CD80⁺CD86⁺ DC cells with CD11c⁺MHC-II⁺ DC cells in the revised manuscript. The statistical
result (n=3) and the gating strategies of DCs (CD45⁺CD11c⁺MHC-II⁺) were shown as follows:

**Figure 5. *In vivo* oncolytic efficacy and immuneactivation capacity of the biomaterialized microbial**
 **nanocomposite.** (a) Schematic illustration of the antitumor activity and immunity investigation of the
 biomaterialized microbial nanocomposite on TC-1-hCD46 xenograft tumor-bearing C57 female mice model. (b)
 Individual tumor growth kinetics in different groups (n=6). (c) The immunofluorescence images of CD8⁺ T cells
 in tumor tissues. Scale bars=50 μm . (d) Representative flow cytometric evolution images (g) as well as relative
 quantification of CD8⁺ T cells (CD45⁺CD3⁺CD8⁺) in the tumor (n=3). (e) Representative flow cytometric
 evolution images (h) as well as relative quantification of Treg cells (CD45⁺CD3⁺CD4⁺Foxp3⁺) in the tumor (n=3).
 (f) Representative flow cytometric evolution images (i) and relative quantification of MHC-II⁺ DC cells
 (CD45⁺CD11c⁺MHC-II⁺) in the tumor (n=3). * $p < 0.05$, ** $p < 0.01$, *** $p < 0.001$, **** $p < 0.0001$ versus control. (G1:

PBS, G2: OMVs@P₂O-Ads, G3: CaP-OMVs-Ads, G4: Intra-Ads, G5: CaP-OMVs@P₂O-Ads, G6: Intra-Ads high
does).

**Question 11:** Discussion section is missing in the current manuscript.

**Response:** We appreciate the reviewer's comments. To help reviewers and readers better
understand the research content of this project, we have included the discussion section in the
revised manuscript (page 22):

“Oncolytic virotherapy is a novel type of immunotherapy inducing antitumor responses through
selective self-replication inside cancer cells and oncolytic virus (OV)-mediated immunostimulation.
It has attracted more attention recently. However, although OVT has incredible advantages in
cancer treatment, the clinical practice of commercial OVs is not perfect. The three oncolytic viral
drugs marketed globally are administered by intratumoral injection. This significantly increases the
difficulty of clinical treatment and decreases medication compliance in patients. In addition, some
clinical trials have attempted to deliver OVs systematically, with unsatisfactory clinical results.

We constructed the microbial nanocomposite for the first time for autophagy-cascade-augmented
immunotherapy. The oncolytic Ads were encapsulated using the engineered OMVs extracted from
*E. coli* and transfected with plasmid to express P₂O. CaP biomineral shells were added to protect
Ads from the clearance of the innate immune system. Therefore, it extends the *in vivo* circulation
time and promotes Ads enrichment after systemic administration. More importantly, P₂O-catalyzed
H₂O₂ elevated the level of oxidative stress in the tumor site, leading to autophagy formation. The
increase in the number of autophagy-induced autophagosomes would significantly augment the
replication efficiency of Ads in OVs-infected cancer cells. Meanwhile, enhanced OVs intratumoral
enrichment augmented OVs replication in tumors and immunosuppressive TEM remodeling based
on the advantage of the immunostimulatory capability of OMVs. This would enhance
OVs-mediated immune responses. Overall, the current autophagy-cascade-boosted immunotherapy
strategy would be promising in OVs-based biomedical therapy applications.”

Minor concerns:

**Question 1:** The language of the paper could be improved with some editing.

**Response:** We appreciate the reviewer's comments. The revised manuscript was checked out

carefully by ourselves and to better improve the readability of the manuscript, we had sent it for
language revision by language revision by Mogoedit language editing service on 23-Feb-2023.

**Figure.** The certificate of MogoEdit language editing services on 23-Feb-2023.

**Question 2:** It would be better to have an introduction of the advantages and disadvantages of
bacterial outer membrane vesicles. Are OMVs better than other nanomaterials? Are there any
potential safety concerns?

**Response:** We appreciate the reviewer's comments. The application of OMVs in the field of drug
delivery has been extensively reported recently ascribed to its intuitive advantages. First, it has
nanoscale particle size, sufficient internal space and wide membrane area to act as a delivery carrier
for a variety of drugs such as Ads. In addition, inheriting various immunostimulatory components
such as lipopolysaccharide (LPS) from their parent bacteria, OMVs also represent a natural immune
activator possessing ability to turn the "cold tumor" into "hot tumor"¹. Furthermore, abandoning
the proliferation ability of the parent bacteria, OMVs has higher controllability and safety than
bacteria². In our subject, engineered bacterial outer membrane vesicles (OMVs@P₂O) are an
irreplaceable component. On the one hand, as mentioned above, serving as the vector for systemic
delivery of the Ads, OMVs@P₂O could protect Ads from recognition and clearance by neutralizing
antibodies. And as the natural immune activator, OMVs@P₂O could remould the suppressive tumor
immune microenvironment for the further oncolytic viral immunotherapy. On the other hand,
OMVs@P₂O naturally carries pyranose oxidase, which could catalyze the production of ROS at the
tumor site and trigger excessive autophagy, thereby improving the replication of Ads in tumor cells.
Overall, OMVs represents an irreplaceable carrier material for the construction of
autophagy-overactivated microbial nanocomposite.

**References**

- 1. Cheng K, et al. Bioengineered bacteria-derived outer membrane vesicles as a versatile antigen
display platform for tumor vaccination via Plug-and-Display technology. Nat Commun 12, 2041
(2021).
- 2. Jahromi LP, Fuhrmann G. Bacterial extracellular vesicles: Understanding biology promotes
applications as nanopharmaceuticals. Adv Drug Deliv Rev 173, 125-140 (2021).

**Question 3:** Figure 1 is missing.

**Response:** We are sorry that Figure 1 had been not shown in the manuscript. We have attached
Figure 1 here and added it to the revised manuscript (page 6).

**Figure 1. Schematic diagram.** The biom mineralized microbial nanocomposite engineered from OV for
 autophagy-cascade-augmented immunotherapy.

**Question 4:** Fig 2g is missing.

**Response:** We appreciate the reviewer's comments. The Figure 2 had been added as follow:

**Figure 2. Preparation and *in vitro* evaluation of the microbial nanocomposite.** (a) TEM and size distribution
 images of Ads, OMVs@P₂O, and OMVs@P₂O-Ads. Scale bar=100 nm. (b) CLSM images of the microbial
 nanocomposite. Ads were stained with DAPI dye (red) and OMVs carried a GFP marker (green). Scale bar=1 μm.
 (c) The expression of P₂O was investigated by the SDS-PAGE method. (d) The ROS level assessment in TC-1

cells by flow cytometry. (e) TEM images of autophagosomes. Scale bar=200 nm. (f) The expression of
autophagy-related protein LC3-I and LC3-II by western bolt analyses. (g) CLSM images of autophagosomes.
Cells were stained with EB dye (red) and autophagosomes were stained with MDC dye (blue). Scale bar=50 μ m.
(h) The Ads replication in TC-1 cells was quantified using real-time PCR at 0, 24, 36, and 48 h sequentially. 3MA
is an autophagy inhibitor: 3-Methyladenine. (i) Cytotoxicity of different formulations in TC-1 cells by CLSM.
Living cells were stained with Calcein (green) and dead cells were stained with PI (red). Scale bar=20 μ m. (j)
Schematic diagram of bridging ROS with oncolytic Ads replication. * p <0.05, ** p <0.01, *** p <0.001,
**** p <0.0001 versus control.

**Question 5:** Fig 2j: What do G1~G6 represent? The authors should mention this information in the
figure legends.

**Response:** We are sorry to make the reviewer confused. The meaning of G1~G6 are as follows: G1:
PBS, G2: Ads, G3: OMVs, G4: OMVs@P2O, G5: OMVs-Ads, G6: OMVs@P2O-Ads. And we
also present this information in the legend in the manuscripts (page 10).

**Question 6:** Fig 2k should be mentioned at least once in the manuscript.

**Response:** We agree with the reviewer's comments. In the revised manuscript, this part was
modified as (page 8): "Autophagy-generated internal double-membrane-bound vesicles
(autophagosomes) could be Ads replication sites within Ads-infected tumor cells. This effectively
enhanced Ads replication through the autophagy pathway (Figure 2j)."

**Question 7:** Fig S7: Only 6 columns are presented, but x-axis has 8 groups.

**Response:** We are sorry for the mistake of marking the number in the post-processing of the Figure
S9. The amendatory Figure S9 was updated as follow:

**Figure S9.** Cytotoxicity of different formulations in TC-1 cells and HCT116 cells by MTT assay. (G1: PBS, G2:
Ads, G3: OMVs, G4: OMVs@P₂O, G5: OMVs-Ads, G6: OMVs@P₂O-Ads).

**Question 8:** Fig S8/5b/6c: What does “Rr=6/6” or “Rr=5/5” mean?

**Response:** We appreciate the reviewer’s comments. “Rr” represents the real survival rate of mice in
each group during pharmacodynamic investigation. “Rr=6/6” or “Rr=5/5” represents that there was
no death of 6 (5) mice in each group (n=6 or n=5) during pharmacodynamic investigation.

**Question 9:** Fig S10: What do G1~G6 represent here. Are they the same with Fig S9? The authors
should mention this information in the figure legends.

**Response:** We are sorry to make the reviewer confused. The meanings of G1~G6 were the same
with Figure S9, namely G1: PBS, G2: Ads, G3: OMVs, G4: OMVs@P₂O, G5: OMVs-Ads, G6:
OMVs@P₂O-Ads. In the revised manuscript, the revisionary legends of Figure S10 was mentioned.

**Question 10:** The authors should explain why they use TC-1-hCD46. Indeed, hCD46 is the receptor
for adenovirus.

**Response:** We appreciate the reviewer’s question. The oncolytic ad11-tel (ad11) was supplied by
Beijing Bio-Targeting Therapeutics Technology Co., Ltd (China). Oncolytic viruses cannot infect
most murine tumor cells because there are no marker molecules on the murine cancer cell surface
that are recognizable to the oncolytic virus. To investigate the use of oncolytic virus in mouse
animal models, we used engineered TC-1-hCD46 murine tumor cell lines by introducing human
CD46 receptor expression plasmid into the murine cancer cells. The oncolytic virus could infect and

replicate in TC-1 cells *via* human CD46 receptors expressed on the cell surface.

**Question 11:** Misspell: “wight” in section 2.2.

**Response:** We appreciate the reviewer’s comments. In the revised manuscript, this misspell was
modified as (page 10): “weight”.

**Question 12:** Fig 3a: The authors ought to give a brief introduction of DiR dye.

**Response:** We agree with the reviewer’s comments. In the revised manuscript, this part was
modified in section 4.11 as (page 27): “DIR is a type of long-chain lipophilic dialkylcarbocyanine
dye. Owing to its lipophilicity, DIR is often used to label cell membranes as well as other
liposoluble biological structures including OMVs. The maximum excitation and emission
wavelengths of DIR are 750nm and 780nm, respectively. Because the infrared light emitted by DiR
can efficiently pass through cells and tissues, it is of great significance in *in vivo* imaging or
tracking.”

**Question 13:** Figure 4: It would be better for the authors to explain why CaP-OMVs exhibit better
tumor selectivity than OMVs.

**Response:** We appreciate the reviewer’s comments. OMVs might still rapidly cause severe
systemic inflammatory response and antibody-mediated clearance. Thus, a "masking" strategy was
adopted in which we used the highly biocompatible calcium phosphate (CaP) to encapsulate OMVs.
Upon the microbial nanocomposite arrival at tumors through EPR effect, the slightly acidic pH of
TME triggered the dissolution of CaP shells, thereby OMVs@P₂O and Ads would be exposed and
play their functions severally.

**Question 14:** Fig 4b: What do C, O, P, S, Ca, and Ca+P stand for?

**Response:** We appreciate the reviewer’s comments. C, O, P, S, Ca, and Ca+P represent element
abbreviations, namely carbon, oxygen, phosphorus, sulphur, calcium and co-localization of calcium
with phosphorus. The revised figure was removed to Figure S14.

**Figure 4. Preparation and *in vivo* evaluation of the biom mineralized microbial nanocomposite.** (a) TEM and
size distribution images of Ads, OMVs@P₂O-Ads, and CaP-OMVs@P₂O-Ads. Scale bar=100 nm. (b) Energy

spectrum analysis image of the biom mineralized composite microbe. Scale bar=50 nm. (c) *In vivo* fluorescence
 imaging of the multiple organs and tumors collected from the mice at 24 h post *i.v.* injection. From left to right:
 tumor, heart, liver, spleen, lung, and kidney. (d) Quantitation of the biodistribution of relative Ads contents in
 multiple organs and tumors after 24 h of different treatments by RT-qPCR (n=4). (e) Immunofluorescence images
 of LC3 autophagic proteins in tumor tissues. Blue represents DAPI-stained tumor cells and the green represents
 FITC-stained LC3 autophagic protein. Scale bar=2 mm. (f) Quantitative analysis of fluorescence intensity. (g) The
 expression of autophagy-related protein LC3-I and LC3-II examined by western blot. (h) Quantitation of relative
 Ads content in the tumor after 72 h of different treatments by RT-qPCR technique (n=3). * $p < 0.05$, ** $p < 0.01$,
 *** $p < 0.001$, **** $p < 0.0001$ versus control. * $p < 0.05$, ** $p < 0.01$, *** $p < 0.001$, **** $p < 0.0001$ versus control.

**Question 15:** Fig S22-23: Gating strategies should also be presented.

**Response:** We appreciate the reviewer's comments. Gating strategies had been presented in Figure
 S30 in the revised manuscript.

**Figure S30.** The gating strategy of effector memory T cells (CD3⁺ CD8⁺ CD62L⁻ CD44⁺) in spleen (n=3). (G1:
 PBS, G2: OMVs@P₂O-Ads, G3: CaP-OMVs-Ads, G4: Intra-Ads, G5: CaP-OMVs@P₂O-Ads, G6: Intra-Ads high
 dose).

**Question 16:** Fig S9/S16/S17/S24/3c/: What do the dotted lines represent?

**Response:** We are sorry to make the reviewer confused. The dotted lines represent error bars here.
 To make the figure information more intuitive for reviewers and readers, we replace the dotted lines
 with the traditional error bars.

**Figure S13.** Body weight changes of TC-1-bearing mice after intratumoral administration of different
 formulations (n=6). (G1: PBS, G2: Ads, G3: OMVs, G4: OMVs@P₂O, G5: OMVs-Ads, G6: OMVs@P₂O-Ads).

**Figure S19.** The tumor of TC-1-bearing mice model volume change for TC-1 xenograft tumor model during
 different treatments (n=6). ****p<0.0001 versus control. (G1: PBS, G2: OMVs@P₂O-Ads, G3: CaP-OMVs-Ads,
 G4: Intra-Ads, G5: CaP-OMVs@P₂O-Ads, G6: Intra-Ads high does).

**Figure S20.** Body weight changes of TC-1-bearing mice after administration of different formulations (n=6). (G1:
 PBS, G2: OMVs@P₂O-Ads, G3: CaP-OMVs-Ads, G4: Intra-Ads, G5: CaP-OMVs@P₂O-Ads, G6: Intra-Ads high
 does).

**Figure S34.** Body weight changes of TC-1-bearing mice after administration of different formulations (n=5). (G1:
PBS, G2: OMVs@P₂O-Ads, G3: CaP-OMVs-Ads, G4: Intra-Ads, G5: CaP-OMVs@P₂O-Ads).

**Figure 3. *In vivo* oncolytic efficacy of the microbial nanocomposite.** (a) *In vivo* DIR fluorescent imaging of the
microbial nanocomposite in TC-1-hCD46 xenograft tumor-bearing mice by IVIS. (b) Schematic illustration of the
antitumor activity and immunology assessment experiments of the microbial nanocomposite using TC-1-hCD46
xenograft tumor-bearing C57 female mice model. TC-1 cells (10^6) were subcutaneously injected into the waist of
female C57 mice, and the tumor-bearing mice were divided into six groups ($n=6$). When the tumor reached
100-150 mm^3 , the mice were injected intratumorally with PBS, Ads (7×10^5 PFU), OMVs, OMVs@P₂O,
OMVs-Ads (7×10^5 PFU), and OMVs@P₂O-Ads (7×10^5 PFU). The drug was given every three days for four
consecutive times, the tumor volume was measured with a vernier caliper, and mice were weighed daily. (c)
**Tumor volume growth profiles of C57 mice bearing TC-1 xenografts.** (d) Images of representative tumors of
different treated groups on the 18th day ($n=6$). (e) Statistical graph of tumor weight of different treated groups
on the 18th day ($n=6$). (f-h) Images of concentration of main cytokines in serum. (i) The differential gene expression
between the samples treated with OMVs@P₂O-Ads and PBS, using the absolute value of logFC greater than 1 as

the threshold. (j) GSEA enriched pathways of the up-regulated genes in the samples treated with
OMVs@P₂O-Ads, showing immune-related terms. (k) Gene set enrichment analysis (GSEA) of the term
“Activation of immune response”, and the genes included in this pathway are highlighted in (i) with light yellow
brown. * $p < 0.05$, ** $p < 0.01$, *** $p < 0.001$, **** $p < 0.0001$ versus control. (G1: PBS, G2: Ads, G3: OMVs, G4:
OMVs@P₂O, G5: OMVs-Ads, G6: OMVs@P₂O-Ads).

**Question 17:** Section 2.5: Some OV_s in clinical trials, including vaccinia virus and reovirus, are
systemically delivered. The authors should mention this and compare the CaP-OMVs technology
with these intravenous OV_s.

**Response:** We appreciate the reviewer’s comments. In the beginning of section 2.5, we mistakenly
said "marketed product" instead of "clinical research stage". In the revised manuscript, this part was
modified in section 2.5 as (page 19): “All marketed OV_s products are delivered by intratumoral or
topical administration rather than intravenous administration.”

We published a review article entitled "Emerging systemic delivery strategies of oncolytic
viruses: A key step toward cancer immunotherapy" in 2021, which introduces the foundation of
clinical trials of OVT to date¹. Although, in clinical trials, there are cases of systemic delivery of
vaccinia virus and reovirus products, it’s still necessary to develop the systemic Ads products due to
its innate advantages. On the one hand, compared with RNA oncolytic virus such as reovirus, Ads
possess higher genetic stability in the process of self-replication in tumor cells, which could ensure
the ability of the virus to infect and kill tumor cells after repetitive replication, thereby maintaining
a low toxicity and high efficiency cancer treatment. On the other hand, compared with vaccinia
virus, Ads have the prospect of product transformation because of lower cost of production.

Herein, constructing the biomaterialized microbial nanocomposite *via* OMVs encapsulation and
biomimetic mineralization technology, we successfully resolved the problem in Ads’ systemic
delivery, including recognition and clearance of neutralizing antibodies, low replication efficiency
in tumor cells and unsatisfactory capacity for immune activation in tumor site. Overall, we firmly
believe that Ads-based systemic delivery possesses the development necessity and we have
provided an ideal strategy in this manuscript.

References

1. Ban W, et al. Emerging systemic delivery strategies of oncolytic viruses: A key step toward

cancer immunotherapy. Nano Res 15, 4137-4153 (2022).

**Question 18:** Section 3: What dose “the oncolytic Ads extracted from *E. coli*” mean? Ads is grown
in HEK293 cells?

**Response:** We are sorry to make the reviewer confused. In the revised manuscript, this part was
modified in section 3 as (page 21): “The oncolytic Ads were encapsulated using the engineered
OMVs extracted from *E. coli* and transfected with plasmid to express P₂O.”

**Reviewer #3:** In the current study, the authors design and develop a modified oncolytic adenovirus
to address the intrinsic drawbacks of the virus. They used biomineral bacterial outer membrane
vesicles encapsulated adenovirus to stimulate autophagy and antitumor immunity. The integrated
immunotherapy is timely and critical for improving the clinical applications of the oncologic
adenovirus and will attract significant attentions from broad readership. There are some important
issues the authors should consider to clarify or improve in the revised version.

**Question 1:** The logic to integrate various components is rather weak and it is recommended for the
authors to clarify in the manuscript. Are these components being replaceable or necessary? It is a
complicated system and it is hardly be treated as composite microbe. It is recommended to change
the word with nanocomposite or nanosystem.

**Response:** We accept the reviewer's proposal to replace "composite microbe" with "microbial
nanocomposite". Oncolytic virotherapy (OVT) is a novel type of immunotherapy that induces
anti-tumor response through selective self-replication within cancer cells and oncolytic virus
(OV)-mediated immunostimulation. However, there are some disadvantages impeding the clinical
practice of commercial OVs, including the poor immune activation capacity and the neutralizing
antibodies elimination. In our study, the engineered OMVs@P₂O was applied to activate anti-tumor
immunity and increasing the replication of Ads in tumor tissue through inducing overactivated
autophagy of tumor cells. Besides, the biomineral calcium phosphate (CaP) shell was used to
protect OMVs@P₂O-Ads from neutralizing antibodies and immune cells. Overall, this Ads delivery
platform described in our manuscript provides the unique insight for clinical applications of
enhanced OVs-mediated cancer immunotherapy, and all of the various components are obligato
and irreplaceable.

**Question 2:** How the adenovirus loaded into OMV? What is the efficacy and any improvement
have been tried?

**Response:** We appreciate the reviewer's questions. In our manuscript, the adenovirus was entered
into the OMV by continuous extrusion through the 200 nm filtration membrane. We demonstrated
that the encapsulation efficiency of OMVs-Ads was more than 90% by fluorescence quenching
experiments with heavy metal ions.

In addition to the extrusion method, we also tried the ultrasound method and the combination of

the two methods to encapsulate adenovirus in OMV. As shown in following Figure, the
 encapsulation efficiency of OMVs-Ads obtained *via* the ultrasound method was lower than the
 extrusion method, and there was no significant improvement observed in the combination group.
 The experimental procedures for fluorescence quenching experiments were as follows. Overall, the
 extrusion method was selected as the most appropriate preparation method of OMVs-Ads in the
 manuscript.

**Figure.** results of fluorescence quenching experiments for OMVs-Ads obtained by different preparation methods.

Method: “Excess Cy7 fluorescent dye was mixed with 10^8 Ads and incubated at 37°C for 3h. And
 free Cy7 was filtered off by ultrafiltration method using a 30kDA ultrafiltration tube. 1mM Cu^{2+}
 solution was configured as fluorescence quenching agent. Ads-Cy7 was divided into 5 parts. The
 extrusion method, the ultrasound method and the combination of the two methods were used
 respectively to encapsulate adenovirus in OMV. $150\ \mu\text{L}$ of bare Ads, bare Ads, and OMVs-Ads
 (extrusion, ultrasound, and combination) were successively added to a black 96-well plate ($n=3$)
 and designated G1 to G5. In G2, G3, G4 and G5, $50\ \mu\text{L}$ of 1mM Cu^{2+} solution was added and
 shaken. The fluorescence intensity of each group was measured by microplate reader (wavelength
 of excitation: 730; wavelength of emission: 770).”

**Question 3:** Autophagy-overactivated is not proper expression, since overactivated action infers to
 uncontrolled process and may lead to severe side effects.

**Response:** We appreciate the reviewer’s comments. The original intention of using
 “Autophagy-overactivated” is to express the following two meanings: One the one hand, there have
 been many studies reported that host tumor cells would autophagy after being infected by Ads.

However, the clinical effect of Ads against tumor cells was not ideal. Herein, we proposed that more
powerful autophagy induced by ROS could enhance intratumoral Ads replication to enhance tumor
killing efficacy. Therefore, to distinguish between the two, we use “autophagy-overactivated” to
refer to the more intense autophagy induced by ROS. On the other hand, the mildly activated
autophagy is a self-protection mechanism for cells to cope with the harsh micro-environment, while
the severe autophagy would lose cyto-protective function and lead to cell death by triggering
autophagic cell death pathway. In our manuscript, we intended to enhance the anti-tumor ability of
Ads *via* autophagy of tumor cells, which required us to induce strong autophagy in tumor cells
instead of mildly activated autophagy. Therefore, to make clear to reviewers and readers the extent
of autophagy referred to in our project, we used the term of “autophagy-overactivated”.

**Question 4:** Quantitative measurement of pyranose oxidase in critical *in vivo*. What is the
contribution for this enzyme for immune activation?

**Response:** We appreciate the reviewer’s kind suggestion. The important role of pyranose oxidase *in*
*vivo* is to promote the generation of ROS in the tumor microenvironment, thereby inducing more
stronger autophagy and enhancing the antitumor efficacy of oncolytic viral immunotherapy.
Therefore, we had evaluated the level of ROS at the tumor site instead of measuring the
concentration of pyranose oxidase *in vivo* (Figure S16).

**Figure S16.** Immunofluorescence images of ROS in tumor tissues. Blue represents DAPI-stained tumor cells and
red represents DHE. Scale bar=100 μ m.

**Question 5:** The scholarly presentation needs to further improve, such as no OV definition provided

in the manuscript.

**Response:** We appreciate the reviewer's comments. We are sorry for the confusion caused to the
reviewer's reading due to the loopholes in scholarly expression. The revised manuscript was
checked out carefully by ourselves and to better improve the readability of the manuscript, we had
sent it for language revision by language revision by MogoEdit language editing service on
23-Feb-2023. Futhermore, we have rechecked the expressive holes in the manuscript, such as no
OV definition provided here, and the revised content has been added in the revised manuscript
(page 4): “An attractive immunotherapeutic strategy is oncolytic viral biotherapy against cancer. It
could selectively kill cancer cells and activate the systemic immune response using oncolytic
viruses (OVs). Oncolytic adenoviruses (Ads) are commonly employed OVs due to their safety and
efficacy.”

CERTIFICATE OF ENGLISH EDITING

This is to certify that the manuscript entitled
**Autophagy-overactivated microbial nanocomposite engineered from
oncolytic adenoviruses for the cascade enhancement of cancer
immunotherapy**
commissioned to us has been carefully edited by a native English-speaking
editor of MogoEdit, and the grammar, spelling, and punctuation have been
verified and corrected where needed. Based on this review, we believe that the
language in this paper meets academic journal requirements. Please contact us
with any questions.

Gang Zhang

Dr. Gang Zhang
Founder & CEO of MogoEdit

Date of Issue
February 22, 2023

Disclaimer: The changes in the document may be accepted or rejected by the authors in their sole discretion after our editing. However, MogoEdit is not responsible for revisions made to the document after our edit on **February 22, 2023**.

MogoEdit is a professional English editing company who provides English language editing, translation, and publication support services to individuals and corporate customers worldwide. As a company invested by the affiliate fund of Chinese Academy of Science, MogoEdit is one of the leading language editing service providers in China, whose clients come from more than 1000 universities and research institutes.

MogoEdit Website: <http://en.mogoedit.com/>
500+ native English editors: <http://en.mogoedit.com/editors>

Mogo Internet Technology Co., LTD.
No. 57, 3rd Keji Road, Xi'an 710075, PR China +86 02988317483 support@mogoedit.com

Figure. The certificate of MogoEdit language editing services on 23-Feb-2023.

**Question 6:** For the immune activation experiments, various critical steps are missing to generate a
concrete conclusion of cascade antitumor activation.

**Response:** We appreciate the reviewer's comments. Oncolytic adenovirus (Ad) was an immune
activation element attracting widespread attention recent years. However, the immune activation

capacity and anti-tumor ability of commercial Ads in clinical stage were unsatisfactory actually.
Herein, to address the the clinical obstacles of Ads, the engineered OMVs@P₂O have been
constructed and introduced in our study. Concretely, when the microbial nanocomposite injected
into the tumor, there would be a plenty of ROS producing through glucose enzymatic hydrolysis by
P₂O, then the excessive accumulation of ROS at the tumor site would sequentially trigger
overactivated autophagy of tumor cells, thereby triggering autophagic immunogenic cells death and
the production of autophagosomes. As reported in relative paper, due to the "imprisonment" effect
of tumor stromal cells on Ads and the rapid death of infected tumor cells, there are no sufficient
condition for the replication of Ads in tumor tissue. Here, a large number of autophagosomes
provide a site for Ads to replicate, and then enhance the Ads-mediated immune response.

In our manuscript, as shown in Figure 2h, this result suggested that the engineered
OMVs-generated ROS could promote autophagy, thereby improving the replication of Ads. After 48
779 hours, compared with OMVs-Ads group and OMVs@P₂O-Ads plus 3MA group (3MA is an
780 autophagy inhibitor: 3-Methyladenine), the Ads replication ability of OMVs@P₂O-Ads was
781 increased by 3.74 ± 0.86 times. And in Figure S14, compared with the group without P₂O (G3 and
782 G5), the agents with P₂O (G4 and G6) remarkably promote the immune activation in the tumor
tissue. Concretely, compared with OMVs group, the proportion of CD8⁺ T cells in OMVs@P₂O
group increased to 1.2 times; compared with OMVs-Ads group, the proportion of CD8⁺ T cells in
OMVs@P₂O-Ads group increased to 1.24 times and compared with PBS group, the proportion of
CD8⁺ T cells in OMVs@P₂O-Ads group increased to 6.15 times. Overall, the introduction of
engineered OMVs@P₂O could draw a concrete conclusion of cascade antitumor activation.

**Figure 2. Preparation and *in vitro* evaluation of the microbial nanocomposite.** (a) TEM and size distribution
 images of Ads, OMVs@P₂O, and OMVs@P₂O-Ads. Scale bar=100 nm. (b) CLSM images of the microbial
 nanocomposite. Ads were stained with DAPI dye (red) and OMVs carried a GFP marker (green). Scale bar=1 μm.
 (c) The expression of P₂O was investigated by the SDS-PAGE method. (d) The ROS level assessment in TC-1

cells by flow cytometry. (e) TEM images of autophagosomes. Scale bar=1 μm . (f) The expression of
autophagy-related protein LC3-I and LC3-II by western bolt analyses. (g) CLSM images of autophagosomes.
Cells were stained with EB dye (red) and autophagosomes were stained with MDC dye (blue). Scale bar=50 μm .
(h) The Ads replication in TC-1 cells was quantified using real-time PCR at 0, 24, 36, and 48 h sequentially. 3MA
is an autophagy inhibitor: 3-Methyladenine. (i) Cytotoxicity of different formulations in TC-1 cells by CLSM.
Living cells were stained with Calcein (green) and dead cells were stained with PI (red). Scale bar=20 μm . (j)
Schematic diagram of bridging ROS with oncolytic Ads replication. * $p<0.05$, ** $p<0.01$, *** $p<0.001$,
**** $p<0.0001$ versus control. G1: PBS, G2: Ads, G3: OMVs, G4: OMVs@P₂O, G5: OMVs-Ads, G6:
OMVs@P₂O-Ads.

**Figure S14.** The infiltration of CD8⁺T cells in tumor of mice treated with different agents on the 18th day.

**Reviewer #4:** This is a meaningful work for the present autophagy-cascade-boosted
immunotherapeutic method. The authors stated that OMVs@P₂O promoted Ads replication and
resulted in Ads-overactivated autophagy, further remolded immunosuppressive TME. However,
several problems that must be clarified need to be solved.

**Question 1:** As we all known, oncolytic adenovirus enters tumor cells through CAR receptor to
play an anti-tumor role. What mechanism does OMVs@P₂O or OMVs@P₂O-Ads enter tumor cells
through? Does it have practical significance in tumor cells with high or low CAR expression?

**Response:** Thanks for reviewer's meaningful question. Human serotype 5 adenovirus (Ad5) is a
non-enveloped virus and its internalization into cells primarily relies on the interaction between
fiber knob of Ad and coxsackie adenovirus receptor (CAR) expressed on cell surface. Once Ad fiber
binds with CAR, a RGD motif at the penton base of Ad interacts with cellular integrin ($\alpha\beta 1$, $\alpha\beta 3$,
or $\alpha\beta 5$) to induce clathrin-mediated Endocytosis. However, in our manuscript, the oncolytic virus
used in this study is Ad11, which relies on CD46 receptor rather than CAR receptor for entry into
cells. In addition, as is reported in related papers, the entry route of OMVs@P₂O or
OMVs@P₂O-Ads is different from that of Ads¹. Concretely, OMVs can bind to certain receptors,
such as Toll-like receptor 2, and activate receptor-induced intracellular signaling in recipient cells.
Besides, OMVs can also be taken up by recipient cells through direct membrane fusion or by using
various endocytic routes, including macropinocytosis, phagocytosis, and endocytosis.

**References**

1. Li M, et al. Bacterial outer membrane vesicles as a platform for biomedical applications: An
update. J Control Release 323, 253-268 (2020).

**Question 2:** The reason of the low intratumoral content of intravenous-delivered Ads is that the
higher level of anti-adenovirus antibody in human body eliminates the exogenous injected Ads. Can
OMVs@P₂O or OMVs@P₂O-Ads effectively avoid the elimination of neutralizing antibodies?
Whether the expression level of anti-adenovirus antibody has been improved in the mouse model in
advance? This is a very necessary experiment.

**Response:** We appreciate the reviewer's comments. The experiment of neutralizing antibody
binding with OMVs@P₂O-Ads has been conducted, and the result image and experimental method

are as follow. As shown in figure, OMVs@P₂O-Ads possess abilities to protect 91.8% Ads from
recognition and clearance by neutralizing antibodies.

Figure. The result of neutralizing antibody binding with OMVs@P₂O-Ads experiment.

**Method:** The serum containing neutralizing antibodies of ads was diluted as 1:100. Then the
diluted neutralizing antibody was mixed with ads or OMVs@P₂O-Ads (10⁷ pfu/mL) and incubated
for 1 h at 4 °C. Afterward, protein G-coated agarose beads (Beyotime, China) were added to the
mixture and incubated for 1 h. The mixture was finally centrifuged at 6000 rpm for 1 min and the
supernatant was collected. The number of ads remaining in the supernatant was determined by
qPCR assay.

**Question 3:** Infection with oncolytic viruses leads to activation of type I IFN signaling pathways,
which are crucial in oncolytic virus-mediated antitumor immunity. The authors stated that
OMVs@P₂O promoted Ads replication. Is this pathway activated to a greater extent by
OMVs@P₂O?

**Response:** We appreciate the reviewer's comments. As suggested by the reviewer, we have
determined the content of type I IFN in the tumor tissue of mice after four different administrations
(G1: PBS, G2: OMVs@P₂O-Ads, G3: CaP-OMVs-Ads, G4: Intra-Ads, G5: CaP-OMVs@P₂O-Ads,
G6: Intra-Ads high does) *via* ELISA experiment. And the experimental result is as follows. As

shown in the figure, the concentration of type I IFN of G5 (CaP-OMVs@P₂O-Ads) was evidently
 higher than G3 (CaP-OMVs-Ads), indicating that the presence of P₂O could enhance the replication
 of Ads.

**Figure.** Images of concentration of type I IFN cytokines in tumor tissue.

**Question 4:** *In vivo* experiment on OMVs@P₂O-Ads or CaP-OMVs@P₂O-Ads regulating tumor
 immune microenvironment is not enough. The innate and adaptive immune cells, as well as the
 activation and exhausted markers of T cells, need to be detected.

**Response:** We appreciate the reviewer's comments. In this manuscript, we have performed a series
 of experiments on the investigation of tumor immune activation. First, to investigate the changes in
 gene expression after OMVs@P₂O-Ads treatments, a transcriptomic analysis of the tumor
 xenografts was conducted to determine the expression of immune-related genes (Figure 3i-k). Then
 we investigated the content of M1-like macrophages (CD45⁺F4/80⁺CD80⁺), (CD45⁺F4/80⁺CD206⁺)
 M2-like macrophages and activated DC (CD45⁺CD11c⁺MHC-II⁺) at the tumor site (Figure S26-S29,
 Figure 5f and 5i). In addition, the amount of CD8⁺ T cells (CD45⁺CD3⁺CD8⁺), IFN-γ⁺CD8⁺ T cells
 (CD45⁺CD3⁺CD8⁺IFN-γ⁺) and Treg cells (CD45⁺CD3⁺CD4⁺Foxp3⁺) were measured (Figure 5c-e,
 g-h, S24 and S25). Furthermore, we performed T cells co-incubation experiment *in vitro* (Figure
 S32) and verified the dependence of CaP-OMVs@P₂O-Ads on CD8⁺ T cells in the process of
 anti-tumor by depleting CD8 T cells with antibodies (Figure S33). Besides, the detection of
 cytokines (Figure 3f-h) in serum and memory T cells (Figure S30 and S31) in spleen can also reflect
 the immune status of tumor to a certain extent.

**Figure 3. *In vivo* oncolytic efficacy of the microbial nanocomposite.** (a) *In vivo* DIR fluorescent imaging of the
microbial nanocomposite in TC-1-hCD46 xenograft tumor-bearing mice by IVIS. (b) Schematic illustration of the
antitumor activity and immunology assessment experiments of the microbial nanocomposite using TC-1-hCD46
xenograft tumor-bearing C57 female mice model. TC-1 cells (10^6) were subcutaneously injected into the waist of
female C57 mice, and the tumor-bearing mice were divided into six groups ($n=6$). When the tumor reached
100-150 mm^3 , the mice were injected intratumorally with PBS, Ads (7×10^5 PFU), OMVs, OMVs@P₂O,
OMVs-Ads (7×10^5 PFU), and OMVs@P₂O-Ads (7×10^5 PFU). The drug was given every three days for four
consecutive times, the tumor volume was measured with a vernier caliper, and mice were weighed daily. (c)
Tumor volume growth profiles of C57 mice bearing TC-1 xenografts. (d) Images of representative tumors of
different treated groups on the 18th day ($n=6$). (e) Statistical graph of tumor weight of different treated groups
on the 18th day ($n=6$). (f-h) Images of concentration of main cytokines in serum. (i) The differential gene expression
between the samples treated with OMVs@P₂O-Ads and PBS, using the absolute value of logFC greater than 1 as

the threshold. (j) GSEA enriched pathways of the up-regulated genes in the samples treated with
 OMVs@P₂O-Ads, showing immune-related terms. (k) Gene set enrichment analysis (GSEA) of the term
 “Activation of immune response”, and the genes included in this pathway are highlighted in (i) with light yellow
 brown. **p*<0.05, ***p*<0.01, ****p*<0.001, *****p*<0.0001 versus control. (G1: PBS, G2: Ads, G3: OMVs, G4:
 OMVs@P₂O, G5: OMVs-Ads, G6: OMVs@P₂O-Ads).

**Figure S26.** Representative flow cytometric evolution images of M1-like macrophages (CD45⁺F4/80⁺CD80⁺) in
 tumor (n=3). (G1: PBS, G2: OMVs@P₂O-Ads, G3: CaP-OMVs-Ads, G4: Intra-Ads, G5: CaP-OMVs@P₂O-Ads,
 G6: Intra-Ads high does)

**Figure S27.** Relative quantification of M1-like macrophages (CD45⁺F4/80⁺CD80⁺) in tumor (n=3). (G1: PBS, G2:
 OMVs@P₂O-Ads, G3: CaP-OMVs-Ads, G4: Intra-Ads, G5: CaP-OMVs@P₂O-Ads, G6: Intra-Ads high does)

**Figure S28.** Representative flow cytometric evolution images of M2-like macrophages (CD45⁺F4/80⁺CD206⁺) in
 tumor (n=3). (G1: PBS, G2: OMVs@P₂O-Ads, G3: CaP-OMVs-Ads, G4: Intra-Ads, G5: CaP-OMVs@P₂O-Ads,
 G6: Intra-Ads high does)

**Figure S29.** Relative quantification of M2-like macrophages (CD45⁺F4/80⁺CD206⁺) in tumor (n=3). (G1: PBS,
 G2: OMVs@P₂O-Ads, G3: CaP-OMVs-Ads, G4: Intra-Ads, G5: CaP-OMVs@P₂O-Ads, G6: Intra-Ads high does)

**Figure 5. *In vivo* oncolytic efficacy and immuneactivation capacity of the biomaterialized microbial**
 **nanocomposite.** (a) Schematic illustration of the antitumor activity and immunity investigation of the
 biomaterialized microbial nanocomposite on TC-1-hCD46 xenograft tumor-bearing C57 female mice model. (b)
 Individual tumor growth kinetics in different groups (n=6). (c) The immunofluorescence images of CD8⁺ T cells
 in tumor tissues. Scale bars=50 μm . (d) Representative flow cytometric evolution images (g) as well as relative
 quantification of CD8⁺ T cells (CD45⁺CD3⁺CD8⁺) in the tumor (n=3). (e) Representative flow cytometric
 evolution images (h) as well as relative quantification of Treg cells (CD45⁺CD3⁺CD4⁺Fcpx3⁺) in the tumor (n=3).
 (f) Representative flow cytometric evolution images (i) and relative quantification of MHC-II⁺ DC cells
 (CD45⁺CD11c⁺MHC-II⁺) in the tumor (n=3). * $p < 0.05$, ** $p < 0.01$, *** $p < 0.001$, **** $p < 0.0001$ versus control. (G1:

PBS, G2: OMVs@P₂O-Ads, G3: CaP-OMVs-Ads, G4: Intra-Ads, G5: CaP-OMVs@P₂O-Ads, G6: Intra-Ads high
does).

**Figure S24.** Representative flow cytometric evolution images of IFN- γ ⁺CD8⁺ T cells (CD45⁺CD3⁺CD8⁺IFN- γ ⁺)
in tumor (n=3). (G1: PBS, G2: OMVs@P₂O-Ads, G3: CaP-OMVs-Ads, G4: Intra-Ads, G5:
CaP-OMVs@P₂O-Ads, G6: Intra-Ads high does)

**Figure S25.** Relative quantification of IFN-γ⁺CD8⁺ T cells (CD45⁺CD3⁺CD8⁺IFN-γ⁺) in tumor (n=3). (G1: PBS,
 G2: OMVs@P₂O-Ads, G3: CaP-OMVs-Ads, G4: Intra-Ads, G5: CaP-OMVs@P₂O-Ads, G6: Intra-Ads high
 does)

**Figure S32.** The experimental result of the co-culture assay. (It's worth noting here that PBS represents T cells
 extracted from mice in the PBS group, and other groups as above.)

**Figure S33.** Tumor volume during the treatments and images of representative tumors of different treated groups
 on the 12th day (n=5).

**Figure S30.** The gating strategy of effector memory T cells (CD3⁺ CD8⁺ CD62L⁻ CD44⁺) in spleen (n=3). (G1:
 PBS, G2: OMVs@P₂O-Ads, G3: CaP-OMVs-Ads, G4: Intra-Ads, G5: CaP-OMVs@P₂O-Ads, G6: Intra-Ads high
 does).

**Figure S31.** Relative quantification of effector memory T cells ($CD3^+ CD8^+ CD62L^- CD44^+$) in spleen (n=3). (G1:
PBS, G2: OMVs@P₂O-Ads, G3: CaP-OMVs-Ads, G4: Intra-Ads, G5: CaP-OMVs@P₂O-Ads, G6: Intra-Ads high
does).

REVIEWERS' COMMENTS

Reviewer #2 (Remarks to the Author):

The authors have addressed all my concerns by conducting additional experiments and analysis. The results are solid.

Reviewer #3 (Remarks to the Author):

Reviewer #4 (Remarks to the Author):

The revised manuscript, "Autophagy-overactivated microbial nanocomposite engineered from oncolytic adenoviruses for the cascade enhancement of cancer immunotherapy" are very effective in addressing all the reviewer's comments and concerns. The manuscript is clearly to be accepted.

Responses to the reviewers' comments

Reviewer #2: The authors have addressed all my concerns by conducting additional experiments and analysis. The results are solid.

Response: Thanks for the reviewer's recognition and support of our work.

Reviewer #4: The revised manuscript, “Autophagy-overactivated microbial nanocomposite engineered from oncolytic adenoviruses for the cascade enhancement of cancer immunotherapy” are very effective in addressing all the reviewer’s comments and concerns. The manuscript is clearly to be accepted.

Response: We are truly grateful to your valuable comments and approval.